# TBK1 activity regulates the directionality of axonal transport of signalling endosomes

David Villarroel-Campos[1,2] , Jose Norberto S Vargas[1,2], Martin Wallace[1], Kai Sun[1,2] , James N Sleigh[1,2] , Pietro Fratta[1,3], Giampietro Schiavo[1,2]

The polarised and complex morphology of neurons poses massive challenges for efficient cargo delivery between the axon and soma, a process termed axonal transport. We have previously shown that the retrograde axonal transport of pro-survival, neurotrophic signalling endosomes relies on Rab7 in motor neurons, and that their trafficking is impaired in the early stages of amyotrophic lateral sclerosis (ALS) pathogenesis. Here, we report the effect of Rab7 phosphorylation on the transport of these signalling endosomes. We show that the ALS-linked kinase TBK1 phosphorylates Rab7 at S72 in neurons, altering its binding to cytoplasmic dynein adaptors. Accordingly, both TBK1 knockdown and the expression of a loss-of-function Rab7 mutant (S72E) induce aberrant bidirectional movement of signalling endosomes without modifying neuronal polarity or endosomal sorting. This alteration is specific for signalling endosomes, as axonal transport of lysosomes and mitochondria remains unaffected. We have therefore discovered a new TBK1 function that ensures the unidirectional transport of signalling endosomes, suggesting that reduced TBK1 activity determines retrograde transport dysfunctions and long-range signalling impairments.

## Introduction

The complex morphology of neurons, and their polarisation into somatodendritic and axonal compartments are essential for their function. The large distance encompassing these domains, which in peripheral neurons can be more than a metre, raises formidable challenges to long-range intracellular communication. For instance, neuronal homeostasis relies on the bidirectional delivery of a variety of cargoes along the axon, a process termed axonal transport, which becomes deregulated in neurological disorders (Sleigh et al, 2019). In particular, target tissues provide neurotrophic factors that are retrogradely transported from axon terminals, and exert pro-survival signalling in the cell body (Scott-

Solomon & Kuruvilla, 2018). This transport occurs in specialised endocytic organelles known as signalling endosomes, which retain signalling competence during their journey back to the soma (Villarroel-Campos et al, 2018). We have shown that brain-derived neurotrophic factor (BDNF) and its receptor tropomyosin receptor kinase B (TrkB) are retrogradely transported in signalling endosomes carrying Rab7A (hereafter referred to as Rab7) (Deinhardt et al, 2006), and that this process is impaired both in vitro and in vivo in genetic models of amyotrophic lateral sclerosis (ALS) (Bilsland et al, 2010; Sleigh et al, 2020; Tosolini et al, 2022).

Rab GTPases are crucial regulators of intracellular trafficking, coordinating the fission, transport, tethering, and fusion of several cytoplasmic compartments (Hutagalung & Novick, 2011). They act as molecular switches, cycling through a GTP-bound active state and a GDP-bound inactive state, promoted by guanine nucleotide exchange factors (GEFs) and GTPase-activating proteins (GAPs), respectively. In addition, Rab function is regulated by post-translational modifications, such as phosphorylation (Homma et al, 2021). In particular, Rab7 is phosphorylated at S72 (p-S72) by TANK-binding kinase 1 (TBK1), leucine-rich repeat kinase 1 (LRRK1), transforming growth factor beta–activated kinase 1 (TAK1), NIMA-related kinase 7 (NEK7), and inhibitor of nuclear factor kappa B kinase subunit epsilon (IKKε) (Heo et al, 2018; Hanafusa et al, 2019; Babur et al, 2020; Ritter et al, 2020; Modica et al, 2025). S72 phosphorylation impairs the interaction between Rab7 and GDP dissociation inhibitor (GDI), and alters the recruitment of effectors (Shinde & Maddika, 2016), with some of them preferentially binding either phosphorylated or unphosphorylated Rab7 (Tudorica et al, 2024), thus impacting on Rab7 cellular functions.

TBK1 is a Ser/Thr kinase involved in the regulation of innate immunity and selective autophagy (Ahmad et al, 2016; Harding et al, 2021). Heterozygous *TBK1* mutations lead to haploinsufficiency and cause ALS (Cirulli et al, 2015; Freischmidt et al, 2015), although the mechanisms linking these mutations to ALS phenotypes remain disputed. Studies carried out in non-neuronal cells have shown that TBK1-dependent Rab7 p-S72 is required for PARKIN-dependent selective mitophagy (Heo et al, 2018), for the regulation of the immune response downstream to stimulator of interferon genes

[1]Department of Neuromuscular Diseases, and UCL Queen Square Motor Neuron Disease Centre, Queen Square Institute of Neurology, University College London, London, UK [2]UK Dementia Research Institute at UCL, London, UK [3]The Francis Crick Institute, London, UK

Correspondence: giampietro.schiavo@ucl.ac.uk

(STING) in breast cancer cells (Ritter et al, 2020), and for relieving mammalian target of rapamycin complex 1 (mTORC1) inhibition in response to amino acid re-feeding in starved cells (Talaia et al, 2024). In motor neurons (MNs), GTP-bound Rab7 is required for axonal transport of signalling endosomes, because expression of a Rab7 dominant-negative mutant severely affects their transport (Deinhardt et al, 2006). Given that the role of Rab7 phosphorylation in the trafficking of signalling endosomes and other cytoplasmic organelles in neurons remains unaddressed, we decided to test whether TBK1 regulates the retrograde transport of signalling endosomes in MNs through Rab7 phosphorylation.

Here, we report a new role of TBK1 in maintaining the unidirectional retrograde movement of signalling endosomes in primary MNs, through the regulation of dynein adaptors recruited by Rab7 p-S72. Because neurotrophic signals use the endocytic pathway to reach the soma, and the same pathway is also impinged by ALS-linked TBK1 deficiency (Shao et al, 2022), we demonstrate a novel function for TBK1 that may contribute to motor neuron pathology.

# Results

## TBK1 loss-of-function impairs the axonal transport of signalling endosomes

ALS-linked *TBK1* mutations are either nonsense or missense mutations, pointing to a loss-of-function pathomechanism (Ye et al, 2019; Gurfinkel et al, 2022). We therefore decided to pursue a knockdown strategy, testing two shRNAs (shTBK1-1 and shTBK1-2) together with a scramble control (shControl) in WT mouse primary motor neuron cultures (Fig S1). MNs transduced with shTBK1-1 or shTBK1-2 showed ~65% knockdown after 72 h, when measured by immunofluorescence, whereas MNs transduced with shControl displayed TBK1 expression levels equivalent to untransduced neurons (Fig S1A and B). TBK1 knockdown is greater when assessed by Western blotting (Fig S1C and D), likely because of the contribution of both neuronal and non-neuronal cells present in the ventral spinal cord cultures. We also confirmed that TBK1 knockdown does not affect neuron viability at 6, 9, and 12 days in vitro (DIV) (Fig S1E and F), in agreement with previous reports (Brenner et al, 2019).

Having validated these two TBK1 shRNAs, we transduced primary MNs on DIV 3 and imaged axonal signalling endosomes 3 d later. For imaging purposes, these organelles were labelled with a non-toxic fragment of tetanus neurotoxin ($H_CT$), which has been shown to undergo axonal retrograde transport in endocytic carriers shared with neurotrophin receptors (Lalli & Schiavo, 2002; Deinhardt et al, 2006). Representative kymographs and displacement graphs for these conditions are shown in Fig 1A and B, respectively. We observed that most cargoes in the untreated and shControl conditions move retrogradely. Although TBK1 knockdown does not alter the percentage of motile signalling endosomes (Fig S1G), we found that a significant proportion of these organelles undergoes anterograde transport under these conditions, irrespective of the shRNA targeting TBK1 used in the experiment (Fig 1B and E). Indeed, the speed

distribution profiles in both the shTBK1-1 and shTBK1-2 conditions are similarly shifted to the left (Fig 1C), reflecting an increase in anterograde transport. Interestingly, the net average speed per cargo remains unaffected (Fig 1D), as is also the case for the run length, percentage of reversal, and time paused (Fig 1F–H); therefore, this phenotype arises from a shift in directionality rather than from a change in the overall cargo speed. As expected, TBK1 knockdown using shTBK1-1 (hereafter called shTBK1) induces an accumulation of signalling endosomes at neurite tips (Fig 1I). Thus, loss of TBK1 in MNs leads to a population of signalling endosomes moving anterogradely, which fail to deliver their pro-survival signals to the soma.

## Rab7 S72E expression causes similar alterations in signalling endosome transport

Because TBK1 loss-of-function modifies the axonal movement of signalling endosomes, we next tested how mutations in Rab7 S72, a residue phosphorylated by TBK1, affect this process. Specifically, we employed the S72A and S72E mutants, because they are tools commonly used to manipulate the Rab7 phospho-state, even though Rab7 S72E does not associate with membranous compartments and behaves as a loss-of-function rather than a phosphomimetic mutant (Heo et al, 2018; Ritter et al, 2020; Talaia et al, 2024). We transfected GFP-Rab7 WT, S72A, or S72E into DIV 5 primary MNs using magnetofection, and imaged signalling endosomes labelled with $H_CT$ on DIV 6. In transfected neurons, we detected two populations of cargoes: (1) endosomes double-positive for Rab7 and $H_CT$, and (2) single-positive endosomes labelled with $H_CT$ alone, except in the Rab7 S72E condition, where just $H_CT$ single-positive cargoes were found, in agreement with the cytoplasmic localisation of the S72E mutant (Fig S2A) (Shinde & Maddika, 2016). Representative kymographs (Fig 2A) and displacement graphs (Fig 2B) show that most cargoes moved towards the soma, except for a subpopulation of signalling endosomes in the Rab7 S72E condition, which were transported anterogradely (Fig 2F). In accordance with the displacement analysis, the speed distribution profiles for double-positive or $H_CT$-only cargoes in MNs expressing Rab7 S72A are indistinguishable from those expressing Rab7 WT (Fig 2C and D), suggesting that additional mechanisms keep anterograde motors inhibited on these cargoes, even upon expression of this Rab7 S72 phosphodeficient mutant. However, single-positive signalling endosomes in MNs expressing Rab S72E display a significant shift to the left (Fig 2D), in agreement with the increase in anterograde transport shown in Fig 2B and F. The average speed per cargo, considering every displacement as positive, shows that cargoes move at similar speeds in every condition (Fig 2E), whereas run lengths, percentage of reversal, and time paused remain also unaltered (Fig 2G–I). Altogether, these results indicate that Rab7 S72E disrupts the axonal transport of signalling endosomes by enabling a directionality switch similar to that observed upon TBK1 knockdown.

## The bidirectional transport of signalling endosomes is not linked to missorting or polarity defects

Because both Rab7 S72E expression and TBK1 knockdown affect the directionality of transport of $H_CT$-positive signalling endosomes, we asked whether this effect is due to alterations of neuronal

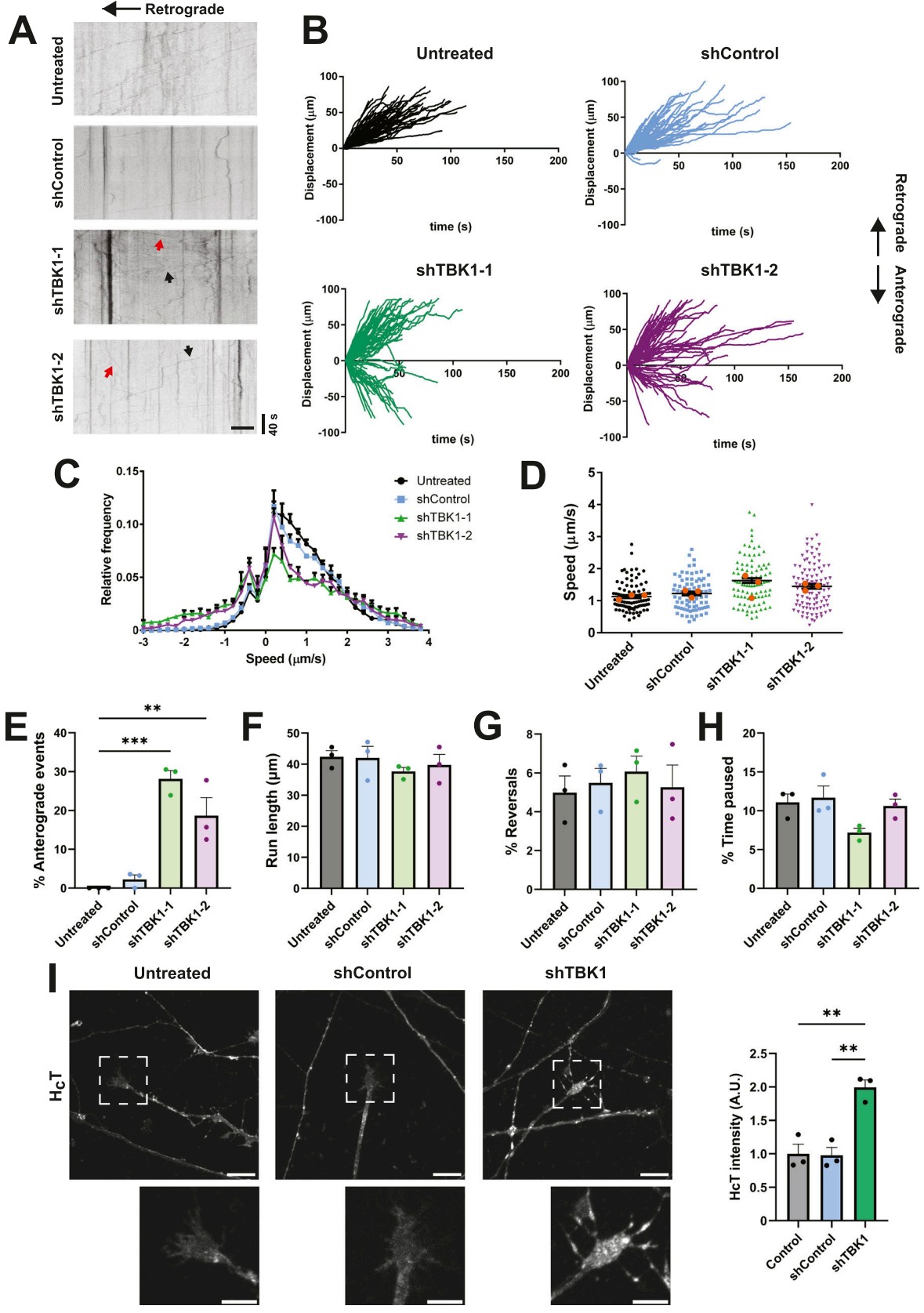

polarity. We magnetofected primary MNs with Rab7 WT or S72E on DIV 2 and subsequently stained them for the somatodendritic marker MAP2 and the axonal marker SMI-31. As shown in Fig S2A, our analysis did not reveal any overt abnormality in neuronal polarity in these neurons. In addition, we magnetofected primary MNs with these Rab7 variants and EB3-mCherry to visualise the polarity of axonal microtubules (Fig S2B); we found that ~98% of microtubules are oriented with their plus ends facing the distal axon in both conditions. To test whether Rab7 S72E impacts the axonal pattern of acetylated (Ac-Tub) and tyrosinated (Tyr-Tub) tubulin, we simultaneously extracted the soluble fraction of the cytoskeleton and fixed MNs expressing Rab7 WT, S72A, or S72E (Fig S2C). MNs expressing the different Rab7 variants exhibit a similar enrichment of Tyr-Tub at the axonal tip, and undistinguishable ratios of Tyr-Tub/Ac-Tub in the proximal and distal axon (Fig S2D). Altogether, these experiments suggest that the observed axonal transport phenotype is not related to changes in neuronal polarity or structural alterations of the microtubule cytoskeleton.

Previous reports suggest that autophagosomes sequentially engage different dynein adaptors during their axonal transport towards the neuronal soma; as a consequence, their otherwise unidirectional movement becomes bidirectional upon maturation through fusion with lysosomes (Maday et al, 2012; Cason et al, 2021). Therefore, we asked whether the observed directionality phenotype may arise from defective sorting of signalling endosomes into lysosomes. We transduced primary MNs on DIV 3 with shControl or shTBK1 and labelled signalling endosomes with H$_C$T and lysosomes with LysoTracker on DIV 6, to quantify double-positive organelles. We found organelles positive for both markers (Fig S2E, arrows) and compartments labelled only with H$_C$T (Fig S2E, arrowheads). Further quantification showed that ~25% of signalling endosomes were also labelled by LysoTracker, a percentage that did not vary amongst treatments (Fig S2F), ruling out missorting to lysosomes as the main cause of bidirectional trafficking.

### TBK1- and Rab7 S72E-dependent bidirectional transport is signalling endosome–specific

Neurotrophin signalling endosomes share common features with late endosomes and lysosomes in spinal MNs, such as the presence of Rab7 in their limiting membrane (Deinhardt et al, 2006). Therefore, we wanted to determine whether the transport disruption caused by TBK1 down-regulation also affects these related cargoes. We thus transduced primary MNs with shControl or shTBK1 on DIV 3, and tracked lysosomes labelled with LysoTracker

on DIV 6. In agreement with previous reports, we found that lysosomes move bidirectionally (Ferguson, 2019), with ~80% of lysosomes moving towards the soma, a percentage that was unchanged across conditions (Fig 3A and B). In addition, TBK1 knockdown has no effect on the speed profile of these organelles (Fig 3C) nor on their average speed (Fig 3D), compared with the untreated condition (Unt).

We then evaluated whether expression of Rab7 WT or the S72 mutants influences the axonal transport of lysosomes. We magnetofected primary MNs in culture with Rab7 WT, S72A, or S72E on DIV 5 and tracked LysoTracker-labelled endosomes on DIV 6. Mirroring the results obtained with TBK1 knockdown, ~80% of lysosomes are transported in the retrograde direction in all conditions (Fig S3A and B), with overlapping speed profiles (Fig S3C) and an unchanged average speed per cargo (Fig S3D).

To further characterise the effects of Rab7 S72 phosphorylation by TBK1 on axonal transport, we decided to monitor the dynamics of a cargo that moves along the axon by a different mechanism. We chose mitochondria, because the recruitment of molecular motors to these organelles depends on the adaptor complex Miro-TRAK1/2 (Mandal & Drerup, 2019), which differs from that of signalling endosomes. We transduced WT primary MNs with shControl or shTBK1 on DIV 3 and tracked mitochondria labelled with tetramethylrhodamine ethyl ester (TMRE) on DIV 6, imaging every 0.5 s. We found that ~75% of mitochondria were static, with the motile fraction equally divided between anterograde and retrograde transport (Fig 3E), a distribution that is not affected by TBK1 knockdown. In addition, we detected no difference in the speed profiles between conditions (Fig 3F), nor changes in the average speed in the anterograde or retrograde directions (Fig 3G and H). We also evaluated possible changes in the axonal distribution of mitochondria (expressed as the number of mitochondria per μm of axon length) and mitochondrial length, finding that both parameters were identical across conditions (Fig 3I–K).

We extended the analysis of mitochondrial axonal transport to MNs expressing Rab7 WT or the S72 mutants. As before, we magnetofected primary MNs with Rab7 WT, S72A, or S72E on DIV 5, and tracked mitochondria labelled with TMRE on DIV 6. In agreement with the results shown in Fig 3, ~75% of organelles are static, and the fraction of moving mitochondria is equally split between anterograde and retrograde motion (Fig S3E). This distribution was not changed by the expression of Rab7 WT or the two mutants. The speed profiles overlap for all conditions (Fig S3F), suggesting that Rab7 WT and the S72 mutants do not alter the axonal transport of mitochondria. In agreement with this, the average speed per cargo

**Figure 1. TBK1 knockdown induces bidirectional movement in a population of signalling endosomes.**
Primary MNs were transduced with lentiviral vectors encoding shControl, shTBK1-1, or shTBK1-2 on DIV 3. After 72 h, cells were labelled with 30 nM Alexa Fluor 647-H$_C$T for 45 min, washed, and imaged. **(A)** Representative kymographs for all conditions, with the cell body located towards the left. In the shTBK1 conditions, black arrows indicate retrograde movement, whereas red arrows show anterograde movement. Scale bar, 10 μm. **(B)** Displacement graphs for all conditions, with retrograde transport shown as positive. A subpopulation of carriers moves anterogradely for both shTBK1-1 and shTBK1-2. **(C)** Speed profiles for each condition; both shTBK1-1 and shTBK1-2 curves are shifted to the left, indicating increased anterograde movement. **(D)** Average speed per cargo, considering all displacement as positive. The average speed is not affected. Number of cargoes: untreated (Unt), 92; shControl, 79; shTBK1-1, 92; shTBK1-2, 103 from three independent cultures; orange dots represent the mean for each replicate; one-way ANOVA with Tukey's multiple comparison test. **(E)** Percentage of anterograde events, considered as endosomes whose position in the last frame is more distal from the soma than their initial position. **(F)** Endosome run length. **(G)** Percentage of reversal movements, measured as changes in direction with a displacement longer than 0.2 μm. **(H)** Percentage of time paused. **(I)** Primary MNs were labelled with 30 nM Alexa Fluor 555-H$_C$T for 45 min, washed, and fixed. Images from neurite tips show an accumulation of H$_C$T at distal regions in the shTBK1 condition. Scale bar, 10 μm in main panels and 5 μm in the insets. For (E, F, G, H, I), n = 3 different cultures; one-way ANOVA with Tukey's multiple comparison test, **P < 0.01, ***P < 0.001.

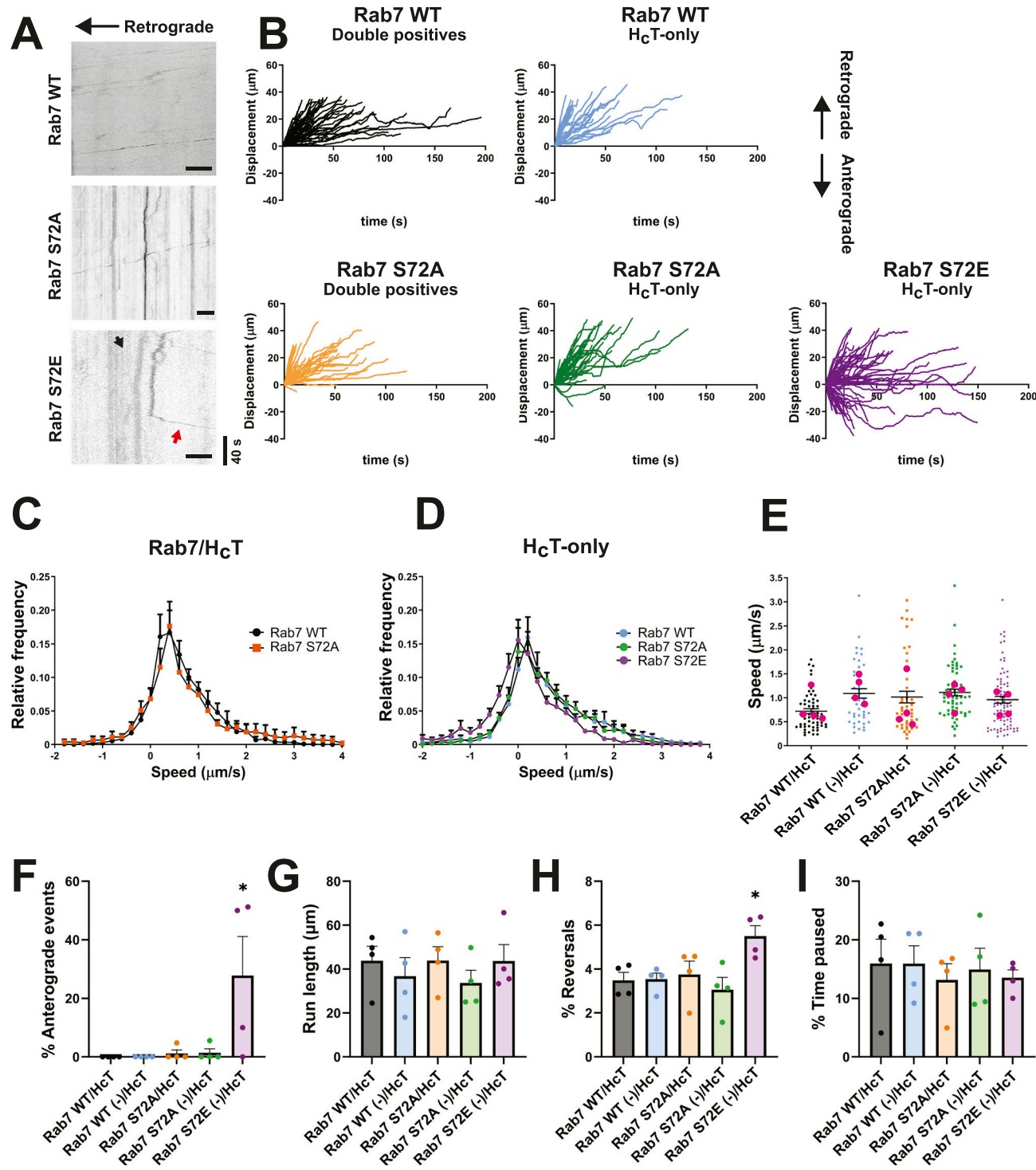

**Figure 2. Rab7 S72E disrupts signalling endosome axonal transport.**
Primary MNs were magnetofected with GFP-Rab7 WT, S72A, or S72E on DIV 5. 24 h later, signalling endosomes were labelled with 30 nM Alexa Fluor 555-H$_C$T for 45 min, washed, and imaged. Cargoes were tracked and classified as double positive (GFP-Rab7 and H$_C$T) or single positive (H$_C$T-only). **(A)** Representative kymographs for all conditions, with the cell body located towards the left. In the S72E condition, the black arrow indicates retrograde movement, whereas the red arrow shows anterograde movement. Scale bar, 10 μm. **(B)** Displacement graphs for every condition, with retrograde movement considered as positive. H$_C$T-containing cargoes move towards the soma; however, a subpopulation of endosomes in the S72E condition moves anterogradely. **(C)** Speed profiles for Rab7/H$_C$T cargoes. Their speed distribution is not changed. **(D)** Speed profiles for single-positive cargoes. The Rab7 S72E curve is shifted to the left, reflecting an increase in anterograde movement. **(E)** Average speed per cargo, considering every displacement as positive. The speed of H$_C$T-positive cargoes is not affected. Number of cargoes: WT/H$_C$T, 59; WT (−)/H$_C$T, 43; S72A/H$_C$T, 45; S72A (−)/H$_C$T, 62; S72E (−)/H$_C$T, 85 from four independent primary cultures; magenta dots represent the mean for each replicate; one-way ANOVA with Tukey's multiple

in the anterograde and retrograde directions was not significantly different (Fig S3G and H, respectively). Mitochondrial distribution remained unchanged across conditions (Fig S3I and J). However, in stark contrast to TBK1 knockdown, we detected an increased proportion of elongated mitochondria upon Rab7 WT and S72A mutant expression (arrows in Fig S3I, quantified in Fig S3K). The cumulative frequency of mitochondrial lengths (Fig S3L) confirms this increased frequency of elongated mitochondria. Altogether, these results rule out the change in axonal transport directionality caused by the loss of TBK1 activity as a global phenomenon, because the dynamics of lysosomes and mitochondria remain largely unaffected.

**Pharmacological modulation of TBK1 activity is cell type–specific**

To further characterise the role of TBK1 in the axonal transport of signalling endosomes, we decided to assess how the activation of this kinase might affect their trafficking. First, we checked whether TBK1 activation induced by polyinosinic:polycytidylic acid (p(I:C))–mediated TLR3 stimulation is conserved in MNs. To this end, we treated primary MNs (DIV 6) with either 10 $\mu$g/ml or 100 $\mu$g/ml p(I:C) for 2 h and measured TBK1 p-S172 levels by Western blotting. Despite being widely used to trigger TLR3 activation and TBK1 phosphorylation in a variety of cellular systems (Czerkies et al, 2018), p(I:C) was ineffective at increasing the phosphorylation of TBK1 at S172 in primary MNs (Fig 4A and B). We also tested whether the TBK1/IKK$\varepsilon$ inhibitor MRT67307 can be effectively used in our cell model. We pre-treated primary MNs with 2 $\mu$M MRT67307 on DIV 6 and stimulated one experimental group with 100 $\mu$g/ml p(I:C) after 1 h, treating instead the other group with vehicle. Surprisingly, cells treated with MRT67307 show a trend towards increased TBK1 activation, which is replicated in N2a cells (Fig 4A and B). p(I:C) did not modify the effect exerted by MRT67307 on TBK1 p-S172 (Fig 4A and B). We also assessed whether p(I:C) was able to activate TBK1, by immunofluorescence. We stimulated primary MNs on DIV 6 with 100 $\mu$g/ml p(I:C) for 2 h and then stained them for TBK1 p-S172. As shown in Fig 4C, p(I:C) treatment does not increase TBK1 activation. We detected, however, a marked accumulation of TBK1 p-S172 near centrosomes in glial cells undergoing cell division (Fig 4C), in agreement with previous reports (Pillai et al, 2015). Because treatment with 10 $\mu$g/ml p(I:C) activates TBK1 in Jurkat cells after 120 min (Fig S4A and B), its inability to do it in MNs indicates the pathway is inactive in this cell type.

Given that Rab7 p-S72 is dephosphorylated by phosphatase and tensin homologue (PTEN), and this in turn regulates the lysosomal targeting and degradation of epidermal growth factor receptor (EGFR) (Shinde & Maddika, 2016), we tested whether TBK1 knockdown could also affect EGFR trafficking. N2a cells were transduced with shTBK1 or shControl, and, 72 h later, stimulated with 100 ng/ml EGF-488 for 20 min and then imaged. We selected the cell outline and then moved it inwards 2 $\mu$m three times, creating four cell profiles

(Johnson et al, 2016), as shown in Fig 4D. TBK1 knockdown reduced EGF-488 overall intensity and significantly decreased the number of EGF-positive organelles located in the most outer ring (Fig 4E and F), suggesting that in addition to neurotrophins, other growth factor signalling pathways are affected by TBK1 knockdown. We also tested how PTEN inhibition may impact on Rab7 localisation in WT MNs, because Rab phosphorylation on their switch II domain alters their binding to GDIs and their membrane/cytosol distribution (Steger et al, 2016). We found that two different inhibitors (bpV [phen] and bpV [HOpic]) increase both the number of Rab7-positive organelles and their Rab7 intensity (Fig 4G and H), suggesting that S72 phosphorylation contributes to maintaining a pool of membrane-bound Rab7.

**Rab7 phosphorylation by TBK1 at S72 controls recruitment of dynein adaptor RILP**

Having shown that TBK1 knockdown induces the bidirectional movement of Rab7-positive signalling endosomes, we sought to identify the mechanism responsible for this phenomenon. First, we purified and tested an antibody against Rab7 p-S72 (Fig S4C and D). Staining with this antibody yielded a punctate distribution in primary MNs, which, as expected, is not changed by p(I:C) stimulation (Fig 5A). Of note, cultured MNs triple-stained for endogenous Rab7, TBK1, and H$_C$T show that 14.2% ± 0.8% (mean ± SEM, n = 3) of H$_C$T-positive endosomes also carry TBK1 and Rab7 (Fig 5B, white arrows). Interestingly, we also observed Rab7/TBK1 particles without H$_C$T, and H$_C$T/TBK1 particles without Rab7 (Fig 5B, blue and yellow arrows respectively, double-positive cargoes quantified in Fig S4E). We carried on confirming TBK1 phosphorylation on Rab7 S72 by an in vitro kinase assay using a library of purified kinases (Table S1) and Rab7 as substrate. To assess the specificity of the phosphorylation, we also used the loss-of-function mutant Rab7 S72P. Although several members of the PKC family phosphorylated both Rab7 WT and the S72P mutant, we found that amongst the kinases tested, only TBK1 and IKK$\varepsilon$ modify Rab7 uniquely at position S72.

Next, we checked whether TBK1 overexpression can increase Rab7 p-S72 levels in HEK-293 cells. We co-transfected FLAG-TBK1 with either GFP-Rab7 WT or S72A and detected a band corresponding to Rab7 p-S72 only in the FLAG-TBK1/GFP-Rab7 WT condition (Fig 5C), confirming that TBK1 phosphorylates Rab7 at S72 both in vitro and ex vivo. In addition, we transfected N2a cells with GFP-Rab7 WT, S72A, or S72E (using GFP alone as control), and isolated the soluble and membrane fractions, using LAMP1 and $\alpha$-tubulin enrichment to confirm the fractionation (Fig 5D). Our results show that the levels of endogenous Rab7 associated with membranes were not changed by the expression of Rab7 WT or either S72 mutants.

Rab7 phosphorylation at S72 alters its binding to effectors (Shinde & Maddika, 2016; Heo et al, 2018; Hanafusa et al, 2019; Tudorica et al, 2024). Hence, we sought to test whether the recruitment of dynein adaptors by Rab7 is also altered upon TBK1-

---

comparison test. **(F)** Percentage of anterograde events. **(G)** Endosome run length. **(H)** Percentage of reversal movements. **(I)** Percentage of time paused. For (F, G, H, I), n = 4 different cultures; one-way ANOVA with Tukey's multiple comparison test. *$P < 0.05$.

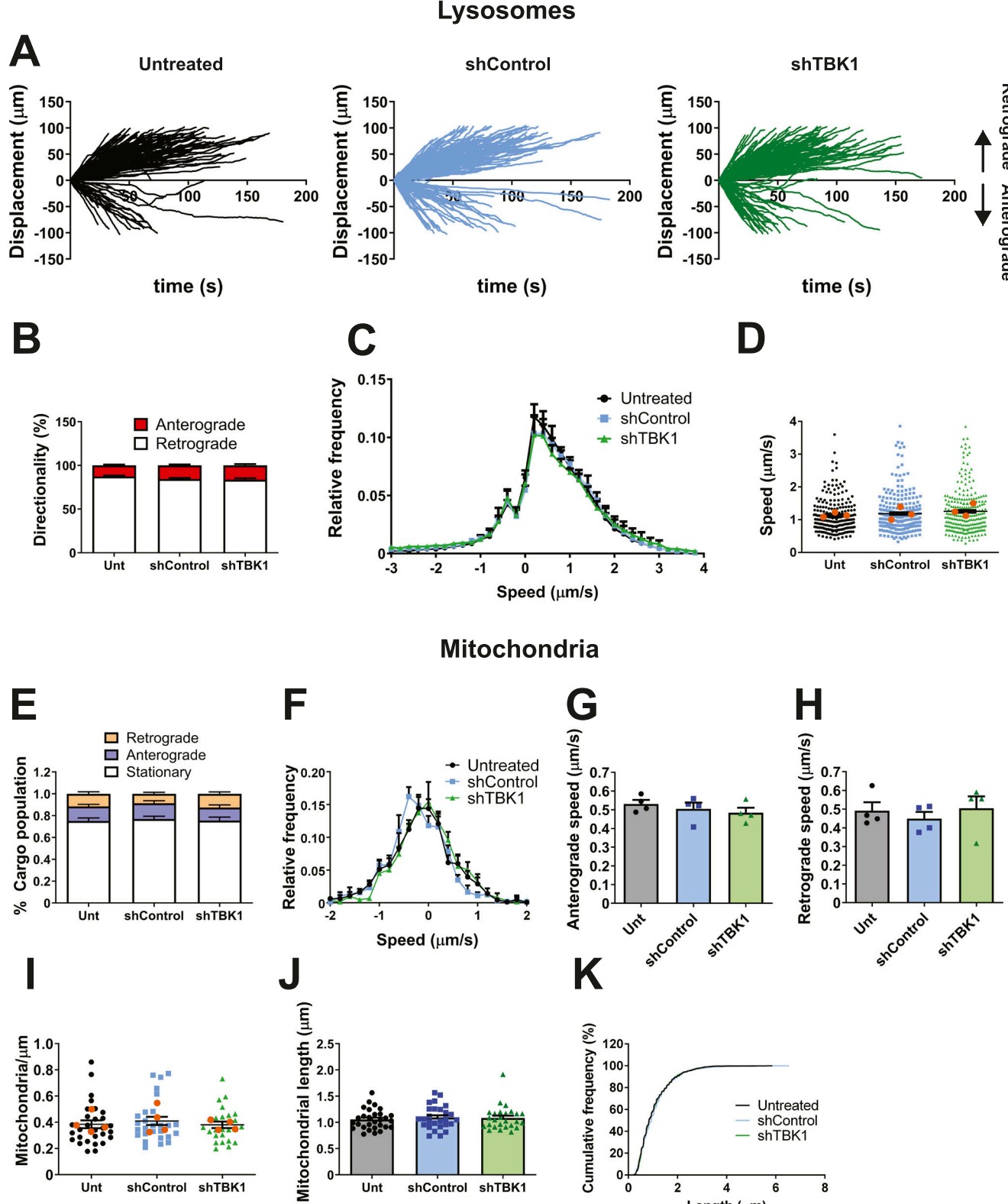

**Figure 3. Axonal transport of lysosomes and mitochondria is unaffected by TBK1 knockdown.**
**(A)** Primary MNs were transduced with either shControl or shTBK1 on DIV 3. 72 h later, MNs were labelled with LysoTracker Deep Red (50 nM) for 30 min, washed, and imaged. The displacement graphs for all conditions show bidirectional movement of cargoes, with a bias towards the retrograde direction. **(B)** Quantification of the directionality of transport. Around 80% of LysoTracker-positive organelles move retrogradely in all conditions. n = 3; two-way ANOVA with Šidak's multiple comparison

dependent phosphorylation. To this end, HEK-293 cells transfected with GFP-Rab7 WT or the S72 mutants were used for GFP-TRAP immunoprecipitations, followed by immunoblotting for Rab-interacting lysosomal protein (RILP) and oxysterol-binding protein-related protein 1L (ORP1L), two established Rab7 effectors that recruit dynein–dynactin to late endosomes and lysosomes (Cantalupo et al, 2001; Johansson et al, 2007), as well as snapin, an adaptor that recruits dynein to TrkB signalling endosomes, mediating their retrograde transport in cortical neurons (Zhou et al, 2012). We found that the expression of Rab7 S72E reduced binding to both RILP and snapin, without affecting ORP1L (Fig 5E). When the GFP-TRAP experiment was performed with lysates prepared from cells overexpressing FLAG-TBK1 and GFP-Rab7, we found that, in contrast to the Rab7 S72E condition, Rab7 S72 phosphorylation increases its interaction with RILP, whereas binding to ORP1L and snapin was unaffected (Fig 5F). This discrepancy between the S72E mutant and TBK1-phosphorylated Rab7 likely arises from the Rab7 S72E cytosolic mislocalisation and consequent loss-of-function (Shinde & Maddika, 2016; Heo et al, 2018).

Altogether, these results show TBK1 ensures the unidirectional transport of neurotrophic signalling endosomes, through Rab7 S72 phosphorylation and the recruitment of the dynein adaptor RILP. Because the axonal transport of other organelles was not affected in our study, this suggests compartment-specific mechanism(s) of motor recruitment modulated by TBK1 in MNs.

# Discussion

Phosphorylation has emerged as a main regulatory mechanism for Rab GTPases, with profound implications for intracellular trafficking deregulation during disease. Here, we report that TBK1-dependent Rab7 p-S72 maintains the unidirectional transport of neurotrophic signalling endosomes in MNs, at least in part through the recruitment of the dynein adaptor RILP. Interestingly, Rab7 S72E overexpression had a similar effect to TBK1 knockdown. However, this mutant exhibits a cytoplasmic mislocalisation, as well as a reduced interaction with components of the geranylgeranyltransferase complex, and with GDP dissociation inhibitors (Shinde & Maddika, 2016; Heo et al, 2018), an abnormal behaviour also reported for Rab8 T72E and Rab10 T73E (Steger et al, 2016). Hence, these variants are predicted to act as a loss-of-function, rather than true phosphomimetic mutants. Nevertheless, because Rab7 S72E expression promoted the anterograde transport of a proportion of signalling endosomes even when the endogenous Rab7 WT was still present, the loss-of-function may in fact correspond to a dominant-negative effect, through the binding to still unidentified aberrant effectors.

Even though Rab7 S72E is unable to model Rab7 phospho-states, we decided to use it as a discovery tool because this mutant has been successfully employed to identify new Rab7 p-S72 interactors, such as folliculin (Heo et al, 2018). Our results show that although the expression of the dominant-negative mutant Rab7 N125I blocks signalling endosome transport (Deinhardt et al, 2006), Rab7 S72E only affects their direction of transport; hence, it is likely to disrupt only Rab7 p-S72–dependent processes.

TBK1 regulates axonal transport directionality in a manner that is specific for signalling endosomes, as lysosomes remain unaffected upon TBK1 down-regulation, despite being Rab7-positive. A similar specificity has been previously observed in signalling endosomes carrying TrkB in hippocampal neurons, whose trafficking relies on the dynein adaptor Hook1 (Olenick et al, 2019). In this model, Hook1 knockdown reduces signalling endosome processivity without affecting the dynamics of LC3B-positive autophagosomes or mitochondria.

A possible explanation for the lack of a global effect of TBK1 on transport might be that this kinase is only present in a sub-population of Rab7 organelles. For instance, a recent report showed occasional TBK1 association with lysosomes in HeLa cells, which increases after TBK1 overexpression (Talaia et al, 2024). Similarly, we found that only 15% of signalling endosomes carry both Rab7 and TBK1 in MNs, pointing towards a fine regulation of TBK1 localisation.

If TBK1 knockdown affects axonal transport globally (e.g., by altering microtubule dynamics in axons or the activity of molecular motors), then organelles unrelated to endosomes, and largely lacking Rab7 on their membranes, like mitochondria, should also be affected. In this regard, our results showed that the axonal transport of mitochondria was unaltered by TBK1 knockdown. This is in agreement with a study analysing ALS-linked TBK1 mutations impairing dimer formation and/or catalytic activity, which found no change in somatic mitochondrial mass upon expression of these mutants in hippocampal neurons in basal conditions (Harding et al, 2021). However, we cannot rule out a possible effect of TBK1 knockdown on mitochondrial homeostasis if mitochondrial function is challenged by depolarising agents. Alternatively, the residual levels of TBK1 after treatment with shRNAs might still be sufficient to ensure some of its functions. We detected, however, an increase in mitochondrial length in MNs expressing Rab7 S72A. Given the established role of Rab7 in mitochondrial fission at mitochondrion–lysosome contact sites (Wong et al, 2018), our results suggest this function might also be modulated by TBK1-dependent phosphorylation of Rab7. Altogether, these observations indicate the regulation of the axonal transport of signalling endosomes is highly specific.

test. **(C)** Speed graphs for every condition, showing overlapping profiles. **(D)** Average speed per cargo, considering every displacement as positive. No statistically significant differences were detected. Number of cargoes: untreated (Unt), 207; shControl, 252; shTBK1, 244 from three independent primary cultures; orange dots represent the mean for each replicate; one-way ANOVA with Tukey's multiple comparison test. **(E)** Primary MNs were transduced with shControl or shTBK1 on DIV 3. 72 h later, MNs were labelled with TMRE (20 nM for 20 min) and imaged. About 75% of mitochondria are static, whereas the rest are split evenly between anterograde and retrograde movement. **(F)** Speed profiles for the three conditions. All profiles overlap almost completely. **(G, H)** Average anterograde and retrograde speed per experiment, respectively. No statistically significant changes were detected. n = 4, one-way ANOVA with Tukey's multiple comparison test. **(I)** Mitochondrial density for each group. No significant changes were detected. Number of mitochondria: Unt, 29; shControl, 27; shTBK1, 26 neurons from four independent cultures; one-way ANOVA with Tukey's multiple comparison test. **(J)** Quantification of mitochondrial length. No significant changes between conditions were detected. Number of mitochondria: Unt, 29; shControl, 27; shTBK1, 26 neurons from four independent cultures; one-way ANOVA with Tukey's multiple comparison test. **(K)** Mitochondrial length cumulative frequency, showing similar distributions for all conditions.

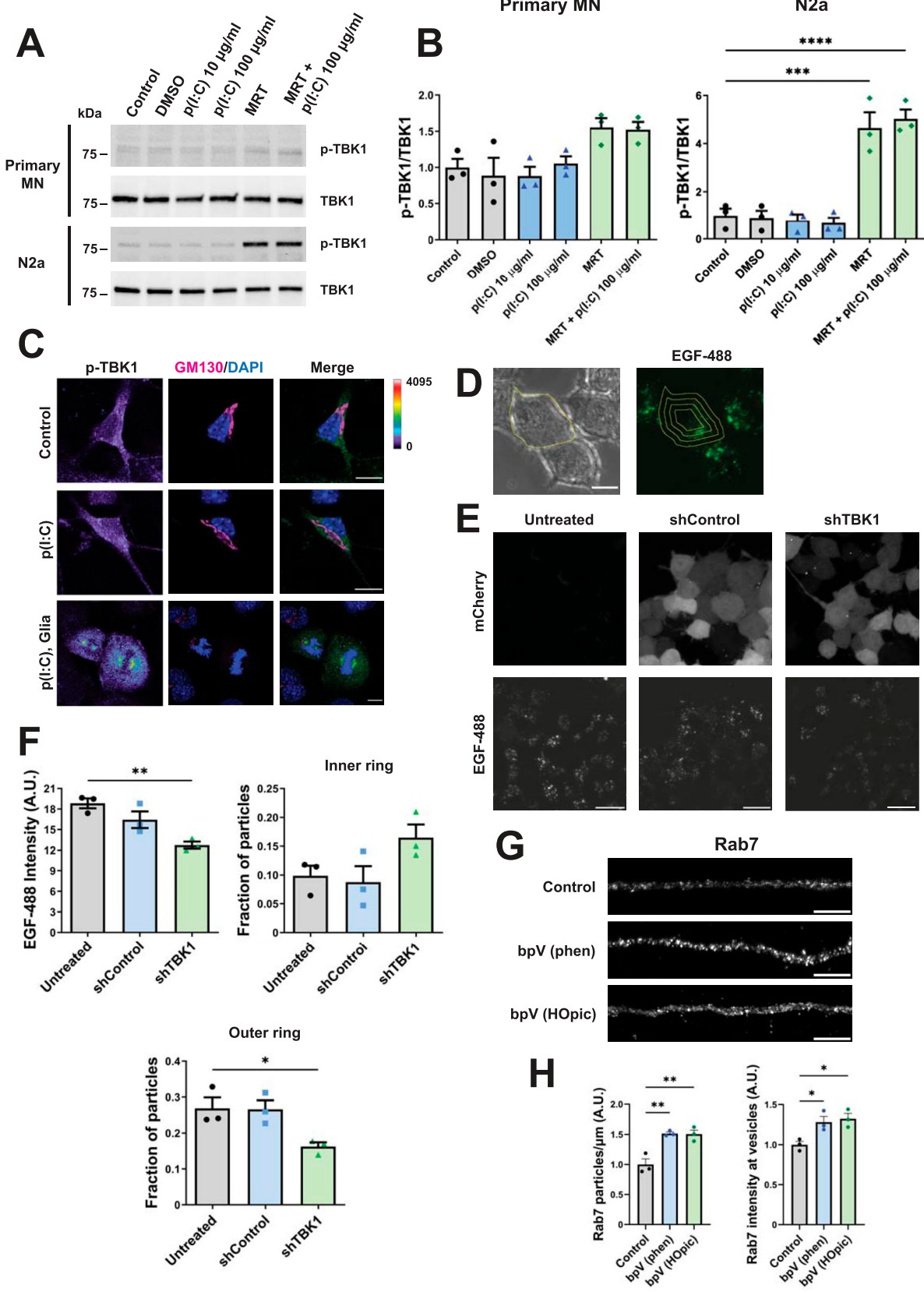

The bidirectional movement of signalling endosomes after TBK1 knockdown points towards an imbalance in the recruitment of molecular motors, which may engage in a "tug-of-war" (Dou et al, 2023). To gain mechanistic insights on this process, we looked for cytoplasmic dynein complex adaptors showing preferential binding for Rab7 p-S72. RILP is one of the best characterised Rab7-binding proteins involved in the recruitment of dynein to lysosomes (Cantalupo et al, 2001; Johansson et al, 2007). However, the effect of Rab7 phosphorylation on RILP binding remains controversial. Conflicting reports have found phosphorylation at S72 increases, decreases, or does not change the strength of Rab7 binding to RILP (Shinde & Maddika, 2016; Hanafusa et al, 2019; Malik et al, 2021). In our experimental conditions, we found that Rab7 p-S72 promotes binding to RILP, whereas the S72E mutants show lower affinity to RILP and negligible binding to snapin, further supporting its loss-of-function characteristics. In contrast, Rab7 S72A still interacts with RILP, in agreement with previous reports (Shinde & Maddika, 2016). This indicates that S72 phosphorylation, rather than being required for RILP binding, has a modulatory role, enhancing or stabilising this association. Henceforth, we propose that TBK1 ensures the unidirectional transport of signalling endosomes by enhancing the recruitment of the retrograde motor dynein through its adaptor RILP to the surface of these organelles. We cannot rule out, however, that TBK1 might act through additional substrates present on signalling endosomes, because its kinase activity specificity is mainly regulated by recruitment to subcellular domains by TBK1 numerous adaptors (Ma et al, 2012; Paul et al, 2025), and in addition, Rab7 S72A expression did not promote signalling endosome anterograde transport. Regarding this, the machinery for anterograde movement also associates with signalling endosomes, given that the proteome of $H_CT$-positive organelles isolated from mouse embryonic stem cell–derived MNs contains the kinesin-1 heavy chains KIF5B and KIF5C, the kinesin light chain 1 and 2, and Arl8b (Debaisieux et al, 2016), a small GTPase having kinesin 1 and 3 as its effectors (Guardia et al, 2016). In this light, we hypothesise that Rab7 S72A fails to alter transport directionality because of a still unidentified anterograde transport inhibition on these organelles, which is relieved by TBK1 knockdown. Therefore, TBK1 and Rab7 p-S72 may function as a toggle mechanism allowing the switch between motors with different directionality.

The modulation of signalling endosome transport directionality exerted by TBK1 presents similarities with the mechanisms by which LRRK2 controls the trafficking of LC3-positive autophagosomes (Boecker et al, 2021), albeit with significant differences. For instance, the axonal transport of autophagosomes is unidirectional in basal conditions but becomes bidirectional upon the expression of the overactive LRRK2 G2019S mutant. In contrast, signalling endosomes move bidirectionally upon TBK1 down-regulation. Furthermore, the retrograde movement of signalling endosomes relies on Rab7, whereas in autophagosomes, LRRK2 acts through a mechanism involving Rab10, ARF6, and the dynein adaptors JIP3 and JIP4 (Cason & Holzbaur, 2023; Dou et al, 2023). In this regard, upon LRRK2 phosphorylation, p-Rab8 and p-Rab10 are able to recruit additional LRRK2 to these organelles, which in turn enhances its kinase activity, establishing a feed-forward mechanism (Vides et al, 2022). Although we currently ignore the molecular determinants recruiting TBK1 to signalling endosomes, it is tempting to speculate the existence of a TBK1/Rab7 p-S72 feed-forward mechanism, which might drive the binding of additional TBK1 onto these organelles.

Because TBK1 loss-of-function alters the axonal transport of signalling endosomes, we attempted to test whether TBK1 activation also has an effect. Unfortunately, TBK1 overexpression was toxic for MNs, whereas established TBK1 pharmacological activators, such as p(I:C), were ineffective in MNs and N2a cells. The latter result is likely to be due to the low expression of the p(I:C) receptor Toll-like receptor 3 (TLR3), in MNs. Despite TLR3 being one of the highest TLRs expressed in this neuronal subtype, its levels are still an order of magnitude lower than in peripheral macrophages (Goethals et al, 2010). Regarding this, it has been shown that interferon-β down-regulates the Rab7 GAP TBC1D15, through the expression of the microRNA miR-1 in *C. elegans* and HeLa cells (Nehammer et al, 2019). This could offer a second mechanism for TBK1-dependent Rab7 regulation, provided that the signalling cascade is active in MN.

Rab7 S72 is also differentially phosphorylated in a cell type–dependent manner. For example, p-S72 can be induced by phorbol 12-myristate 13-acetate (PMA) in mouse embryonic fibroblasts, but not in the human skin keratinocyte cell line HaCaT (Malik et al, 2021). Conversely, HaCaT cells show increased Rab7 p-S72 levels after treatment with EGF (Malik et al, 2021), highlighting the importance of studying this signalling cascade in the cell type of interest. Unexpectedly, we found that the established TBK1 inhibitor MRT67307 activates TBK1 in N2a cells and to a

**Figure 4. Pharmacological modulation of TBK1 activity is cell type–specific.**
**(A)** Primary MNs (upper panels) were stimulated for 2 h with 10 or 100 µg/ml p(I:C) on DIV 6. Another group was treated with 2 µM MRT67307 for 3 h, or 2 µm MRT67307 for 1 h and 100 µg/ml p(I:C) for another 2 h in the presence of the inhibitor. TBK1 p-S172 and total TBK1 levels were determined by WB. The same conditions were used to stimulate undifferentiated N2a cells (bottom panels). p(I:C) fails to activate TBK1, whereas the TBK1 inhibitor MRT67307 enhances the levels of active TBK1. **(B)** Quantification of TBK1 p-S172/TBK1 from primary MNs (left panel) and N2a cells (right panel) from **(A)**. n = 3, one-way ANOVA with Tukey's multiple comparison test. ***$P < 0.001$, ****$P < 0.0001$. **(C)** Primary MNs were stimulated with 100 µg/ml p(I:C) for 2 h on DIV 6, fixed, and stained as indicated. TBK1 p-S172 fluorescence intensity levels are shown with a rainbow scale. p(I:C) did not change TBK1 p-S172 levels or its subcellular distribution. An example of non-neuronal cells undergoing mitosis in the same primary culture is presented at the bottom. Centrosomes show clear accumulation of TBK1 p-S172. Scale bar, 10 µm. **(D)** N2a cells were transduced with shControl or shTBK1, stimulated 72 h later with 100 ng/ml EGF-488 for 20 min, and imaged. The cell outline was drawn and eroded inwards 2 µm three times, creating four concentric rings. Scale bar, 10 µm. **(E)** Representative images of EGF-488 uptake, with mCherry expression as the transduction reporter. Scale bar, 20 µm. **(F)** EGF-488 intensity quantification across the whole cell, as well as fraction of particles in the inner or outer ring. TBK1 knockdown decreases the overall EGF signal in N2a cells, whilst also reducing the number of EGF-positive organelles in the cell periphery. n = 3, one-way ANOVA with Tukey's multiple comparison test, *$P < 0.05$, **$P < 0.01$. **(G)** Primary MNs were treated on DIV 6 with 100 nM bpV (phen) or bpV (HOpic) for 1 h, followed by fixation and endogenous Rab7 immunolabelling. Scale bar, 5 µm. **(H)** Quantification of the number of Rab7 puncta/µm, as well as Rab7 intensity in those puncta. PTEN inhibition increases both Rab7 organelle density and Rab7 vesicular fluorescence intensity. n = 3, one-way ANOVA with Tukey's multiple comparison test, *$P < 0.05$, **$P < 0.01$.
Source data are available for this figure.

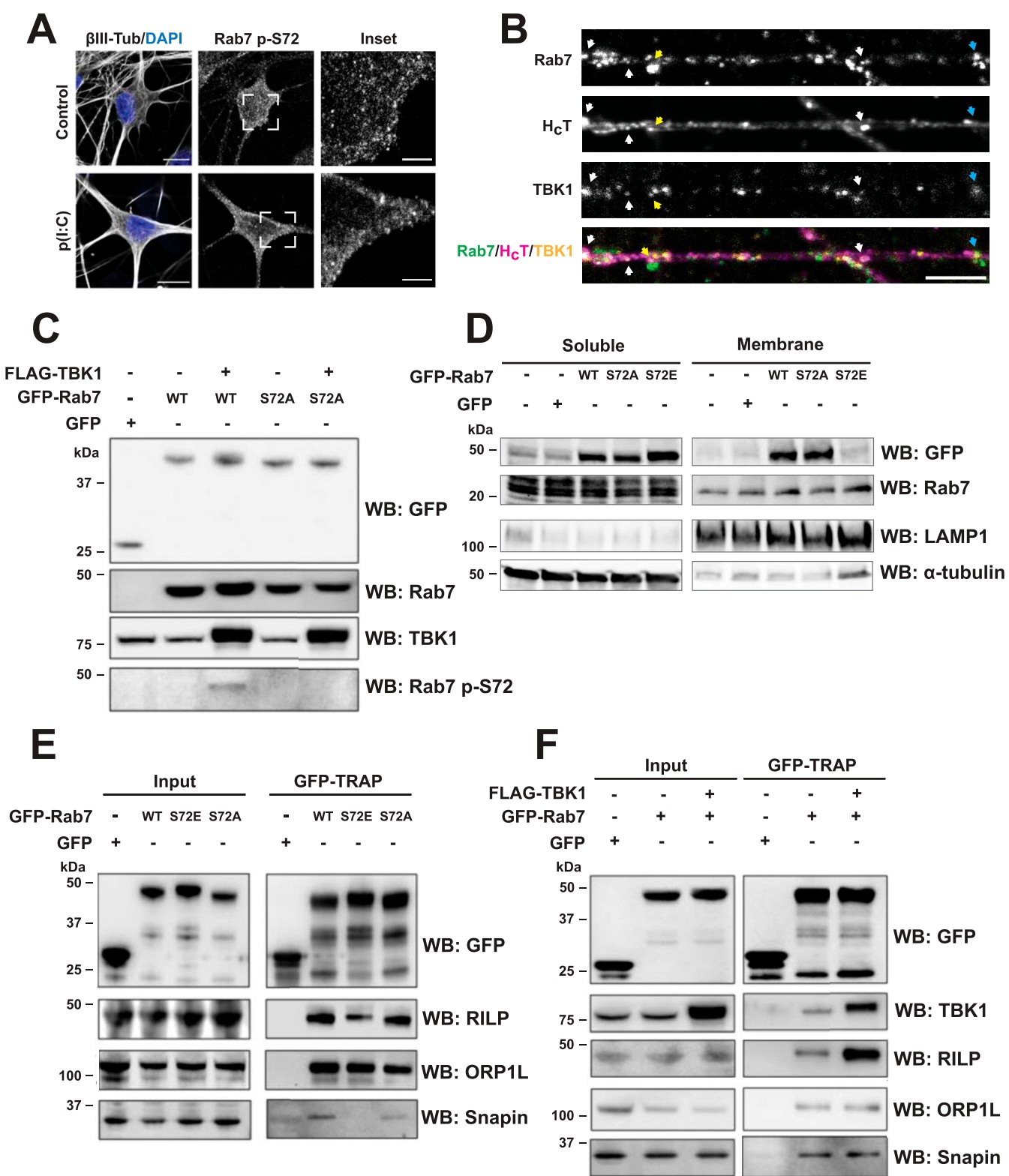

**Figure 5. TBK1-dependent Rab7 p-S72 regulates binding to the dynein adaptor RILP.**
**(A)** Primary MNs were treated on DIV 6 with 100 μg/ml p(I:C) for 2 h, fixed, and stained as indicated. p(I:C) did not alter Rab7 p-S72 levels or its punctate distribution. Scale bar, 10 μm. Insets show examples of Rab7 p-S72 puncta. Scale bar, 3 μm. **(B)** 6 DIV primary MNs were labelled with 30 nM Alexa Fluor 555-H$_C$T for 45 min, washed, and fixed. Double staining for endogenous Rab7 and TBK1 allows to detect triple-positive cargoes for Rab7/TBK1/H$_C$T (white arrows), as well as Rab7/TBK1 organelles (blue arrow) and TBK1/H$_C$T carriers (yellow arrow). Scale bar, 5 μm. **(C)** HEK-293 cells were co-transfected with FLAG-TBK1 and either GFP-Rab7 WT or S72A (GFP alone was used as a control). The anti-Rab7 p-S72 antibody recognises a band only in the TBK1/Rab7 WT condition. **(D)** N2a cells were transfected with GFP-Rab7 WT or the

lesser extent in MNs. Other studies have shown that the TBK1 inhibitors BX795, MRT67307, and GSK8612 do not completely prevent TBK1 trans-autophosphorylation at p-S172, hinting at the existence of upstream kinases activating TBK1 (Ye et al, 2022 Preprint).

Given the deficits in axonal transport of signalling endosomes observed in ALS models (Sleigh et al, 2020; Tosolini et al, 2022), what could be the contribution of TBK1 loss-of-function to disease pathogenesis? TBK1 deficiency has been linked to endolysosomal dysfunction. $TBK1^{-/-}$ human MNs exhibit impaired endosomal maturation and reduced lysosomal function, which results in TAR DNA-binding protein 43 (TDP-43) mislocalisation to the cytoplasm (Hao et al, 2021 Preprint). This effect can also be triggered by poly(GA) derived from C9orf72 ($G_4C_2$) expansion, with poly(GA) inclusions sequestering TBK1, which in turn induces early endosome enlargement and TDP-43 aggregation, a phenotype that is worsened by the ALS-linked TBK1 R228H mutation (Shao et al, 2022). In addition, mice homozygous for the ALS-causing mutation TBK1 E696K show an enrichment of enlarged lysosomes lacking degradative enzymes in spinal cord MNs, pointing towards a disrupted lysosomal biogenesis and/or turnover (Brenner et al, 2024). Thus, the abnormal bidirectional transport of signalling endosomes could be another manifestation of how this pathway is affected by TBK1 loss-of-function, specifically in MNs.

TBK1 exhibits a biphasic function during ALS pathology, being protective at early stages in a cell-autonomous manner, but detrimental at later stages, mainly because of non–cell-autonomous mechanisms related to its role in inflammation (Brenner et al, 2019). We hypothesise that reduced TBK1 activity compromises the delivery of pro-survival signals carried by signalling endosomes, impacting long-term motor neuron survival, and fitting with the protective role of TBK1 at disease onset. As the reversion in transport is partial, it is possible that the negative effect on survival only appears at a late stage, or it requires a "second hit," in line with recent TBK1 ALS models exhibiting mild but steady accumulation of subtle cellular phenotypes (Brenner et al, 2024). It is also possible that these survival phenotypes may only become overt in a compartmentalised culture system, where trophic factors are exclusively present at distal axons, instead of our mass culture approach. This set-up would also rule out any possible somatic $H_CT$ endocytosis event that could lead to missorting into anterogradely transported carriers.

In summary, we report a novel neuronal role of TBK1 in controlling the directionality of axonal transport of neurotrophic signalling endosomes partially through the phosphorylation of Rab7 at S72 and the recruitment of the dynein adaptor RILP. This effect is specific for these organelles, highlighting a new mechanism for differential axonal transport regulation of distinct cargoes in MNs.

# Materials and Methods

## Animals

All experiments were carried out following the guidelines of the Queen Square Institute of Neurology Genetic Manipulation and Ethic Committees, and in accordance with the European Community Council Directive of 24 November 1986 (86/609/EEC). Animal experiments were undertaken under license from the UK Home Office in accordance with the Animals (Scientific Procedures) Act 1986 (Amended Regulations 2012). WT mice from a C57BL/6 x SJL background were used in all experiments.

## Antibodies and reagents

In this work, we used the following antibodies: anti-acetylated tubulin (sc-23950, 1:400; Santa Cruz), anti-$\alpha$ tubulin (ab6161, 1:2,000; Abcam), anti-$\beta$III-tubulin (Tuj1) (MMS-435P, 1:1,000; Covance), anti-GM130 (#610822, 1:100; BD Transduction laboratories), anti-GFP (sc-9996, 1:2,500; Santa Cruz), anti-LAMP1 (sc-5570, 1:1,000; Santa Cruz), anti-MAP2 (AB5622, 1:1,000; Chemicon), anti-ORP1L (R247, a gift from Vesa Olkkonen, 1:1,000), anti-TBK1 p-S172 (mAb#5483; Cell Signaling, 1:500 in Western blot [WB], 1:200 in immunofluorescence [IF]), anti-Rab7 (ab50533; Abcam, 1:1,000 in WB, 1:250 in IF), anti-RFP (A00682, 1:1,000; GenScript), anti-RILP (ab140188, 1:1,000; Abcam), anti-SMI-31 (SMI-31R, 1:1,000; Covance), anti-snapin (148,002, 1:3,000; Synaptic Systems), anti-TBK1 (ab40676; Abcam, 1:3,000 in WB, 1:200 in IF), and anti-tyrosinated tubulin (mab1864, 1:500; Millipore). LysoTracker Red DND-99, LysoTracker Deep Red, and TMRE (T669; 1 mM stock) were purchased from Thermo Fisher Scientific. Poly(I:C) LMW (p(I:C)) and EGF-488 were from Invitrogen, and bpV (phen), bpV (HOpic), and MRT67307 were obtained from Sigma-Aldrich.

## Anti-Rab7 p-S72 antibody generation and purification

A peptide comprising the sequence flanking Rab7 S72 was synthesised with S72 phosphorylated (CERFQpSLGVA-CONH$_2$) and used for the immunisation of two rabbits by BioGenes GmbH. For antibody purification, phosphorylated and unphosphorylated forms of the peptide were dissolved in coupling buffer (50 mM Tris–HCl, pH 8.5, 5 mM EDTA), added to SulfoLink Resin (Thermo Fisher Scientific), and incubated for 1 h. Excess peptide was removed and the reaction stopped by adding coupling buffer containing 50 mM cysteine (pH 8.0) for 30 min. Resins were packed into separate columns and washed sequentially with water, wash buffer 2 (1 M NaCl, 200 mM glycine, pH 2.4), and water at 4°C. The hyperimmune serum was diluted 1:1 in Tris-buffered saline (TBS: 50 mM Tris–HCl, pH 7.4, 150 mM NaCl), added to the unphosphorylated peptide column, and incubated at 4°C for 2 h under rotation. The flow-through was then added to the phosphorylated

---

S72 mutants (including GFP as a control) for 24 h, after which they were used to obtain membrane protein–enriched fractions. LAMP1 and $\alpha$-tubulin were chosen to monitor the membrane fraction enrichment. Endogenous Rab7 levels in the membrane fraction are not affected by expression of GFP-Rab7 WT, nor by the S72 mutants. **(E)** GFP-TRAP from HEK-293 cells transfected with GFP-Rab7 WT, S72A, or S72E as indicated (GFP alone as a control). Rab7 S72E binding to both RILP and snapin is decreased, compared with Rab7 WT and S72A. **(F)** GFP-TRAP from HEK-293 cells transfected with FLAG-TBK1 and GFP-Rab7 WT as indicated (GFP alone as a control). WB shows RILP, ORP1L, and snapin co-precipitation with Rab7. TBK1 expression increases Rab7/RILP interaction.

peptide column and incubated at 4°C for 2 h under rotation. The column was washed twice with ice-cold TBS, and the antibody was eluted in 200 mM glycine, pH 2.4. The eluate was immediately neutralised with 1.5 M Tris–HCl, pH 8.8, and bovine serum albumin (BSA) was added to 0.1% final concentration. After purification, the antibody was tested by ELISA, as follows: phosphorylated and unphosphorylated peptides were diluted in TBS, 0.01–10 ng added to a 96-well plate and allowed to air-dry overnight. The next day, wells were blocked with TBS containing 0.1% Tween-20 and 1% BSA for 1 h at RT. The purified antibody was incubated for 2 h, wells were washed three times, and an anti-rabbit secondary antibody conjugated to HRP was added for 1 h. Wells were then washed five times before addition of SIGMAFAST OPD substrate (P9187; Sigma-Aldrich) for 10 min. The reaction was stopped by adding 50 $\mu$l 2 M HCl, and the absorbance was measured at 450 nm.

## Plasmids

Canine Rab7 WT cloned into the mammalian expression vector pEGFF-C1 was kindly provided by Cecilia Bucci (Bucci et al, 2000). Rab7 S72A and S72E were generated by site-directed mutagenesis (QuikChange Site-Directed Mutagenesis Kit, Stratagene). EB3-mCherry was a gift from Michael Davidson (plasmid # 55037; Addgene). A shRNA set directed against TBK1 plus a control sequence were purchased from GeneCopoeia (# MSH101526-LVRU6MP), with an U6 promoter and mCherry as a reporter gene. Control sequence: GCTTCGCGCCGTAGTCTTA, shRNA 1: CCAGAATCA GAATTTCTCATT, shRNA 2: CCAGTTCTTGCAAACATACTT.

## In vitro kinase assays

The in vitro kinase validation was carried out by ProQinase GmbH, using a radiometric protein kinase filter–binding assay and a library of 190 serine/threonine kinases, using His$_6$-Rab7 WT or S72P as substrates. In vitro phosphorylation of GST-Rab7 by TBK1 (Thermo Fisher Scientific) was performed in reaction buffer (20 mM Tris–HCl, pH 7.5, 10 mM MgCl$_2$, 1 mM EGTA, 10 $\mu$M Na$_3$VO$_4$, 0.5 mM $\beta$-glycerophosphate, 2 mM dithiothreitol [DTT], and 0.01% Triton X-100) with 100 $\mu$M ATP and 100 ng of TBK1 per 50 $\mu$l of reaction. Samples were incubated at 30°C for 30–60 min, boiled, and analysed by Western blotting.

## Cell culture

Tissue culture reagents were purchased from Thermo Fisher Scientific, unless stated otherwise. N2a and HEK cells were maintained in DMEM supplemented with 10% FBS and 1% GlutaMAX at 37°C and 5% CO$_2$. Jurkat cells were grown in RPMI media with 10% FBS at 37°C and 5% CO$_2$. Cells were routinely tested for *Mycoplasma* using a MycoAlert detection kit (Lonza). Primary MNs were isolated as previously described (Arce et al, 1999; Lalli & Schiavo, 2002). MNs were resuspended in motor neuron medium (Neurobasal, 2% heat-inactivated horse serum, 2% B27 supplement, 1X GlutaMAX, 25 $\mu$M 2-mercaptoethanol, 1% penicillin/streptomycin, recombinant rat CNTF [10 ng/ml; R&D Systems], recombinant rat GDNF [0.1 ng/ml; R&D Systems], and recombinant human BDNF [1 ng/ml; R&D Systems]). Magnetofection (NeuroMag, OZ Biosciences) was carried

out following the manufacturer's instructions (with 0.5 $\mu$g of DNA, 1.75 $\mu$l beads, and 15 min of incubation on top of a permanent magnet). Magnetofected neurons were incubated for 18–24 h and used for further experiments.

## Western blotting

For WB, cells were lysed in RIPA buffer (50 mM Tris–HCl, pH 7.5, 1 mM EDTA, 2 mM EGTA, 150 mM NaCl, 1% NP-40, 0.5% sodium deoxy-cholate, and 0.1% sodium dodecyl sulphate containing protease and phosphatase inhibitors [HALT, Thermo Fisher Scientific]), for 30 min at 4°C. Lysates were centrifuged at 21,000$g$ for 15 min at 4°C, and protein concentration was determined. Between 10 and 20 $\mu$g of total protein extract was loaded on 4–15% Mini-PROTEAN TGX precast gels (Bio-Rad) or NuPAGE 4–12% Bis-Tris gels (Thermo Fisher Scientific) and transferred into polyvinylidene difluoride (PVDF) membranes (Bio-Rad). Membranes were blocked in TBS containing 0.05% Tween-20 and 5% BSA for 1 h at RT and then incubated with primary antibodies overnight at 4°C. Membranes were washed and incubated with HRP-conjugated secondary antibodies (GE Healthcare) for 1 h at RT. Immunoreactivity was detected using Immobilon Classico ECL Substrate (Millipore) and ChemiDoc MP Imaging System (Bio-Rad).

## Membrane fractionation

N2a cells were transfected with GFP-Rab7 WT, S72A, or S72E using Lipofectamine 3000 (Thermo Fisher Scientific) in Opti-MEM (Gibco) at 70% confluency for 24 h. Cells were then used for the purification of soluble/membrane-enriched protein extracts, with the Mem-PER Plus Membrane Protein Extraction Kit (Thermo Fisher Scientific), following the manufacturer's instructions. The fractions enriched in cytosolic or membrane proteins were used to determine Rab7 levels by WB.

## Immunoprecipitation

GFP-TRAP (ChromoTek) immunoprecipitations were performed using HEK-293 cells transfected with Lipofectamine 3000 in Opti-MEM at 70% confluency for 24 or 48 h. Cells were lysed in TBS lysis buffer (10 mM Tris–HCl, pH 7.5, 150 mM NaCl, 0.5% Nonidet P-40 Substitute) containing HALT protease and phosphatase inhibitors by pipetting and keeping them on ice for 30 min. Cell lysates were then centrifuged for 10 min at 15,000$g$ at 4°C; the supernatant was collected and then incubated with equilibrated GFP-TRAP beads at 4°C for 1 h under rotation. Beads were then washed with TBS three times in a magnetic rack. Bound proteins were eluted from the GFP-TRAP beads using NuPAGE 2X LDS sample buffer (Thermo Fisher Scientific) with 100 mM DTT (Sigma-Aldrich) at 99°C. Supernatants were separated from the beads using a magnetic rack, and samples were then processed for WB.

## Immunofluorescence

Cells were fixed in 4% PFA and 4% sucrose in PBS for 30 min at RT, washed three times with PBS, permeabilised with Triton X-100 0.2% in PBS for 5 min, washed again three times, and blocked in 5% BSA

in PBS for 1 h at RT. Primary antibodies were incubated overnight at 4°C in blocking solution. Coverslips were washed three times, incubated with Alexa Fluor–conjugated secondary antibodies for 1 h at RT, washed again three times, stained with DAPI when indicated, washed once with water, and mounted on Mowiol. TBK1 localisation experiments in MNs were carried out in the same way, except that cells were permeabilised with saponin 0.05% in PBS for 5 min, and blocked with 5% BSA, 0.05% saponin in PBS for 1 h. Primary and secondary antibodies were incubated in the same blocking solution. For the tyrosinated/acetylated tubulin analysis, MNs were magnetofected on DIV 2. After 24 h, neurons were simultaneously extracted and fixed, as previously described (Ahmad et al, 2000). Briefly, MNs were washed with PBS and then extracted/fixed with a solution containing PHEM (60 mM PIPES-NaOH, 25 mM Hepes-NaOH, 10 mM EGTA, and 2 mM $MgCl_2$, pH 6.9), 4% PFA, 0.15% glutaraldehyde, and 0.2% Triton X-100 for 15 min. After this, the immunofluorescence protocol was carried out as described above.

### Immunofluorescence quantification

Fluorescence intensity profiles for both tyrosinated and acetylated tubulin were analysed with Fiji/ImageJ (Schindelin et al, 2012). Profiles were measured starting at the axonal tip towards the cell body, until obtaining a profile of at least 50 $\mu m$. The ratio between both channels on the proximal and distal axon was determined as average measurement from segments of 5 $\mu m$ in length, between 5 and 10 $\mu m$ from the axon tip (shown as distal), and between 45 and 50 $\mu m$ from the axonal tip (shown as proximal). For the analysis of $H_CT$ accumulation at neurite tips, we measured the mean fluorescence intensity in the last 10 $\mu m$ of each distal process.

For the EGF-488 endosome subcellular distribution experiment, we outlined the border of each cell analysed, and then enlarged the selection by –2 $\mu m$ (three times), as reported previously (Johnson et al, 2016), before calculating the fraction of particles in the inner or outer ring.

For the determination of Rab7 fluorescence associated with vesicles in control neurons, or after PTEN inhibition, we created a mask for Rab7-positive particles with a threshold above the Rab7 cytosolic signal, using Fiji/ImageJ, followed by counting number of particles and measuring fluorescence intensity in those particles. Similarly, to quantify the number of Rab7/$H_CT$/TBK1 endosomes, we identified $H_CT$-positive structures and checked what percentages were also positive for the other two endogenous fluorescence signals.

### Live-cell imaging and tracking

Primary MNs were transduced on DIV 3 or magnetofected on DIV 5, as described above. On DIV 6, neurons were labelled with either 30 nM Alexa Fluor–conjugated $H_CT$ for 45 min, 50 nM LysoTracker Red or Deep Red for 30 min, or 20 nM TMRE for 20 min. Cells were washed and imaged in motor neuron medium supplemented with 20 mM Hepes-NaOH, pH 7.4 (for mitochondrial transport experiments, the imaging medium also contained 20 nM TMRE). Images were acquired every 0.5 s, for a total of 400 images per time lapse (300 frames for mitochondrial experiments). Endosomes and

lysosomes were tracked using the Fiji/ImageJ plugin TrackMate (Tinevez et al, 2017) in manual mode. The proportion of motile/stalling cargoes was determined from kymographs generated with the Fiji/ImageJ Multi Kymograph tool. Only endosomes moving for at least 10 frames were tracked, until they exited the imaging window or reached a terminal pausing, which corresponds to the absence of movement for more than 10 frames. Mitochondria were tracked using the ImageJ plugin KymoAnalyzer (Neumann et al, 2017) with the following parameters: pixel size: 0.1 $\mu m$, frame rate: 2 frames/s, multiplication factor in "Segments" plugin (plugin 5): 2, threshold for the detection of mobile/stationary and switching tracks: 0.5 $\mu m$, and threshold for the assignment of segments/pauses: 0.1. For the EB3 comet experiments, MNs were magnetofected on DIV 2 and imaged after 6 h. Images were acquired every 0.5 s, for a total of 100 images per time lapse. Kymographs for the EB3 comet experiments were also generated with the Multi Kymographs analysis option in Fiji/ImageJ.

### Statistical analysis

Statistical analysis was performed using GraphPad Prism software. Data are shown as the mean ± SEM. We compared multiple groups by one-way ANOVA with Tukey's multiple comparison test and grouped conditions by two-way ANOVA with Šidak's multiple comparison test. For cases when variances were non-comparable, the Kruskal–Wallis non-parametric one-way ANOVA was used. We used the D'Agostino–Pearson test as a normality test. Statistical significance was considered as follows: *$P < 0.05$, **$P < 0.01$, ***$P < 0.001$, and ****$P < 0.0001$.

### Supplementary material

Fig S1 presents the validation of the TBK1 knockdown and its effect on primary MN survival, as well as the proportion of motile endosomes in the TBK1 knockdown and Rab7 magnetofection experiments. Fig S2 rules out polarity defects or changes in endosomal sorting as the cause of altered axonal transport. Fig S3 shows that Rab7 S72E does not impact the axonal transport of lysosomes or mitochondria. Fig S4 demonstrates that TBK1 activation by p(I:C) is cell type–specific. In addition, it presents the validation of our Rab7 p-S72 antibody and reports the percentage of double-positive cargoes for the Rab7, $H_CT$, and TBK1 pairs. Table S1 reports the kinases tested in the Ser/Thr kinase screen.

# Supplementary Information

# Acknowledgements

We thank members of the Molecular NeuroPathobiology (MNP) and Greensmith laboratories (UCL Queen Square Institute of Neurology) for critical reading of the article and personnel of the Denny Brown Laboratory

(UCL) for assistance with mouse colonies. We also thank Dr. Anna-Leigh Brown, Dr. Riccardo Zenezini Chiozzi, and Dr. Konstantinos Thalassinos for their help with sample and data analyses. This work was supported by a MNDA PhD Fellowship number 880-792 (D Villarroel-Campos), a CONICYT PhD Scholarship 2016/72170645 (D Villarroel-Campos), Brain Research UK Miriam Marks Fellowship in Neurodegeneration (BF-100029) (JNS Vargas), Target ALS Springboard Fellowship (FS-2023-SBF-S2) (JNS Vargas), Medical Research Council awards (MR/S006990/1 and MR/Y010949/1) (JN Sleigh), the Wellcome Trust Senior Investigator Awards (107116/Z/15/Z and 223022/Z/21/Z) (G Schiavo), the Korean Ministry of Trade, Industry and Energy (MOTIE) and the Korea Institute for Advancement of Technology (KIAT) support through the 'International Cooperative R&D Program' (Task No. P0028350) (G Schiavo), and the UK Dementia Research Institute award (UKDRI-1005) (G Schiavo). The authors declare no competing financial interests. All data required to evaluate the conclusions in the article are present in the article and/or in the Supplementary Materials.

## Author Contributions

D Villarroel-Campos: conceptualisation, resources, formal analysis, validation, investigation, visualisation, project administration, and writing—original draft, review, and editing.
JNS Vargas: conceptualisation, formal analysis, validation, investigation, and visualisation.
M Wallace: formal analysis and investigation.
K Sun: investigation.
JN Sleigh: formal analysis, funding acquisition, and investigation.
P Fratta: conceptualisation and funding acquisition.
G Schiavo: conceptualisation, resources, supervision, funding acquisition, project administration, and writing—original draft, review, and editing.

## Conflict of Interest Statement

The authors declare that they have no conflict of interest.

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
