## [Reviewer comments · Life Science Alliance]

TBK1 activity regulates the directionality of axonal transport of signalling endosomes

David Villarroel-Campos, Jose Vargas, Martin Wallace, Kai Sun, James Sleight, Pietro Fratta, and Giampietro Schiavo
DOI: <https://doi.org/10.26508/lsa.202503527>

Corresponding author(s): Giampietro Schiavo, University College London

Review Timeline:	Submission Date:	2025-12-12
	Editorial Decision:	2026-01-05
	Revision Received:	2026-01-13
	Editorial Decision:	2026-01-15
	Revision Received:	2026-01-21
	Accepted:	2026-01-21

Scientific Editor: Tim Fessenden

Transaction Report:

Please note that the manuscript was previously reviewed at another journal and the reports were taken into account in the decision-making process at *Life Science Alliance*.

Reviews

Reviewer #1

Comments to the Authors (Required):

This paper reports the consequences of TBK1 depletion on axonal transport of signaling endosomes in motor neurons (MNs). In general the experiments are clearly explained but given prior work by others, there are some flaws in the logic and issues of impact that need to be addressed before publication in this journal can be recommended.

The authors are studying phosphorylation of Rab7 S72 which is well established to be phosphorylated by additional kinases including LRRK1, IKK ϵ , TAK1, and NEK7. As reported by others (Hanafusa et al., 2019), phosphoRab7 shows higher affinity for RILP protein compared with non-phosphoRab7 (Fig. 5 F). If the S72E mutant is a true phospho-mimetic mutation, it should also show higher affinity for this protein but it does not—it binds more poorly (Fig. 5E) and it shows cytoplasmic localization (as discussed by the authors). Such incorrect behavior for Rab phosphorylation mimics has been seen by others for this Rab and Rab8 and Rab10—such mutant proteins cannot be used to reflect {plus minus}phosphorylation states. Because the data indicate strongly that these are non-functional proteins, interpreting the consequences of their expression is not straightforward nor helpful.

First the authors compare anterograde and retrograde trafficking and see loss of retrograde in the absence of TBK1. This is the expected result if pRab7 recruits RILP to recruit dynein (as shown by others previously). (Please include % events anterograde vs. retrograde in all conditions, not just speed). Next they study S72E which cannot be interpreted and should be removed because it cannot be explained at this stage (see above). The authors then show that the transport defects are specific to sorting endosomes and not acidic lysosomes (Lysotracker+) or mitochondria—this would be expected since those compartments do not have much Rab7 (please state this). The authors report that TBK1 is not activated by p(I:C) in MNs. They also show that knockdown of the pRab7 phosphatase, PTEN, appears to increase retrograde transport of EGF containing endosomes, as reported by others.

Finally, they generated a pRab7 antibody (although a highly validated one is commercially available from Abcam). They tested an incomplete library of kinases and missed LRRK1, an established Rab7 kinase not mentioned herein. They showed that overexpressed TBK1 phosphorylates Rab7 in cells. Mass spec of cell extracts failed to identify known pRab7 effectors and most hits were below the level of significance, suggesting that the protocol used was suboptimal and probably too preliminary for presentation at this stage; (without proper validation, the data should be presented elsewhere).

Putting it all together, despite a great deal of work, the overall novelty of the story is low and unfortunately does not significantly advance our understanding of endosomal retrograde transport compared with prior work in this area. The authors stress a role for RILP in the discussion as part of what they have shown, but they did not really include RILP analysis in their transport experiments. Does loss of RILP phenocopy loss of TBK? In summary, the area of this work is interesting but the overall impact of this manuscript is limited. In future work, the authors might explore relative expression of Rab7 kinases in MNs versus other cell types to better understand TBK1's unique contribution to endosomal retrograde transport. That could possibly raise the impact of this story—also, looking at total phosphoproteins in these cells {plus minus} TBK1 to understand what is regulating its activity.

Other comments:

Page 5 line 12 "Rab7 GTPase activity" is misleading as hydrolysis is not needed...it would be more correct to state: GTP-bound Rab7 is needed

Page 6. Line 9. If the culture is contaminated with glial cells, they would also contribute to the western blot. More likely is the use of a non-linear western blot method that exaggerates signal to noise—fluorescent antibodies give a more linear and reliable response than HRP amplification.

Page 12. Line 4. Not clear how this result " suggest S72 phosphorylation contributes to membrane targeting and localization of Rab7"? PhosphoRabs are a small fraction of total Rabs in cells, and phosphorylation happens after targeting and localization.

Reviewer #2

Comments to the Authors (Required):

In this manuscript, the authors address the role of the kinase TBK1 in regulating axonal transport. Mutations in TBK1 can lead to ALS, and this kinase has been shown to phosphorylate Rab7, a GTP-ase involved in vesicle sorting and trafficking. The authors have a very interesting finding that knockdown of TBK1 selectively alters transport of tetanus-toxin containing endosomes, the signaling endosomes, and makes the transport bidirectional rather than exclusively retrograde. However, the mechanism for this change is not clear from the study. In addition, there are a number of issues with individual experiments and conclusions.

1. In the analyses of motility (Figures 1,2), the authors do not include data on the percent of vesicles that are stationary, nor do they evaluate run duration, pauses or reversals. These should all be evident from the kymographs, and would be helpful to know in order to interpret the effects of TBK1 on motility.
2. The authors exclusively use tetanus toxin to investigate transport of signaling endosomes. While it is clear that the signaling endosomes containing neurotrophic factors and signaling components also contain tetanus toxin, I do not think that all of these tetanus toxin containing endosomes also contain neurotrophic signals. This comes up particularly when the authors say (p.6) "TBK1 knockdown does not affect neuron viability at 6, 9 and 12 DIV" and yet. "loss of TBK1 in MINs leads to a population of signalling endosomes moving anterogradely, which fail to deliver their pro-survival signals to the soma." The system used does not make the MNs rely on signaling endosomes for support, as they would if trophic factors were exclusively presented to distal axons and were being transported in this way.
3. The changes observed in mitochondrial length are interesting, and there are a large number of possible explanations. Additional alternatives would be helpful, as would an experiment to determine whether there are changes in mitochondrial function (ie TMRE staining).
4. The data provided indicate that, surprisingly, MRT67307 actually increases activation of TBK1. Given the emphasis of the paper on determining the effects of TBK1 activation on transport of signaling endosomes, it seems surprising that the authors did not investigate the effect of this drug on transport.
5. Similarly, the data indicate that bpV (HOpic), a PTEN inhibitor, alters the "number of Rab-7 positive organelles and their intensity", and the authors indicate that this is due to changes in Rab7 phosphorylation. If this is the case, which would be interesting, does bpV actually alter Rab7 phosphorylation in this system, and does that affect transport?
6. The authors carry out mass-spec analyses to identify proteins that preferentially interact with phosphorylated Rab7. The data show that the one consistent and statistically significant interactor was TBK1. However, this is hard to interpret, as the phosphorylated Rab7 was generated by incubating Rab7 with TBK1, and so the presence of TBK1 is likely to be an artifact. If the authors want to make any statements about TBK1, they should repeat these experiments carrying out the phosphorylation in vitro with another kinase such as IKK. The main point that the author try to make from the mass spec data is an interaction with dynein heavy chain. This was not statistically significant in the data, although the authors state "we sought to confirm the enhanced dynein binding for Rab7-pS72". But the authors do not carry out additional experiments to show that there is, in fact, any association between dynein heavy chain and Rab7. Instead, they look at dynein interactors known to be involved in Rab7 functions. Taken together, the mass spec data do not seem to be helpful for the study.
7. In figure 5F, the experiments to show enhanced pulldown of RILP when TBK1 and Rab7 are overexpressed should include a control in which GFP alone is expressed with TBK1, and then GFP is pulled down and the pulldown is blotted with RILP. As it is, TBK1 expression could increase non-specific pulldown of RILP. It would also be good for the experiments in Figure 5C to include a kinase dead TBK1, to show that this does not lead to the p-S72 Rab7 signal.

Minor issues

1. The authors indicate that 14.2% of HcT-positive endosome also carry TBK1 and Rab7. It would be interesting to provide the number for all the colocalization analysis of these three markers.
2. The authors state "These results show TBK1 ensures the unidirectional transport of neurotrophic signalling endosomes, through Rab7 S2 phosphorylation and the recruitment of cytoplasmic dynein. However, the data do not show any evidence of recruitment of dynein."

Reviewer #3

Comments to the Authors (Required):

ALS is a progressive neurodegenerative disorder caused by the death of upper and lower motor neurons. Genetic mutations in dozens of genes have been implicated in ALS, including heterozygous loss-of-function mutations in TBK1, a Ser/Thr kinase involved in various cellular processes. In this study, the authors demonstrate that: 1, TBK1

knockdown disrupts the axonal trafficking of signaling endosomes but not other organelles; 2, TBK1 phosphorylates Rab7 at Ser72 in neurons, and expression of a Rab7 mutant (S72E), which models a loss-of-function state, also impairs signaling endosome transport, suggesting that Rab7 mediates TBK1's function in this process; and 3, phosphorylation of Rab7 by TBK1 at Ser72 regulates recruitment of the dynein adaptor RILP. These observations represent a limited conceptual advance in our understanding of ALS pathogenesis and TBK1 biology for the following reasons. 1. Although it is useful to report the observation that TBK1 knockdown disrupts a specific aspect of endosome biology-namely, its axonal trafficking-the regulation of endosomes by TBK1, as well as its impairment in ALS, has been extensively documented (e.g., EMBO Rep., 2025; Nat. Cell Biol., 2023; Science, 2022; EMBO Rep., 2019; J. Immunol., 2011). Similarly, defects in axonal transport in ALS have been studied for decades. 2. Rab7 is a well-established phosphorylation target of TBK1 especially at Ser72 (e.g., EMBO J., 2024; Sci. Adv., 2018). 3. RILP is a known Rab7-interacting protein. Thus, this study provides quite limited novel insights into TBK1 biology and the contribution of partial TBK1 loss of function to ALS.

The authors also report that p(I:C), a compound widely used to induce TBK1 phosphorylation in various cell types, did not affect TBK1 phosphorylation in primary motor neurons (Fig. 4). While this is a potentially interesting observation, without further mechanistic investigations, it does not add much to this study thus should not be presented as a regular figure in the paper.

Main comments

1. Inclusion of additional parameters from the kymographs as indicated by Reviewer 2, point 1 and for all % events anterograde vs. retrograde in all conditions (Reviewer 1):

In the analyses of motility (Figures 1,2), the authors do not include data on the percent of vesicles that are stationary, nor do they evaluate run duration, pauses or reversals. These should all be evident from the kymographs, and would be helpful to know in order to interpret the effects of TBK1 on motility. (R2)

Please include % events anterograde vs. retrograde in all conditions, not just speed. (R1)

We thank the Reviewers for raising this point, as these transport parameters may offer additional insights into the changes induced by TBK1 knockdown and Rab7 S72E expression. We now added to the manuscript the percentage of anterograde events, run length, percentage of reversal and the percentage of time that endosomes paused, both for TBK1 knockdown in **Figure 1E-H** and Rab7 overexpression in **Figure 2F-I**. Regarding the percent of vesicles that are stationary, we opted to present the combined analysis for TBK1 and Rab7 by generating kymographs for all conditions assessed in Figures 1 and 2 and displaying the percentage of motile cargoes in **Supplementary Figure 1G**.

2. Removal of results describing use of M.S. to identify pRab7 effectors following concerns raised by Reviewers 1 and 2 on the preliminary nature of this experiment.

We were disappointed to learn that the Reviewers recommend the exclusion of these data, given that one of our main goals was addressing the cell-type specificity of TBK1 activity, and that, for the first time as far as we are aware, we report about the detection of Rab7 interactors in embryonic spinal cord tissue. At the same time, we fully acknowledge the lack of classical Rab7 interactors and the absence of dynein complex components (a recurrent problem in mass spec analysis looking at dynein adaptors) in our analyses. Following the recommendation of the Editor and the Reviewers, we have therefore removed the mass spec results from **Supplementary Figure 4** and in the text.

3. Expansion of the discussion section for observations on (a) mitochondrial trafficking, and (b) effect of TBK1 knockdown on viability in the cultured neurons in terms of requirement of trophic factors, and (c) phenotypes of S27E mutant in the context of its affinity for RILP and cytoplasmic localisation as elaborated by Reviewer 1:

(A) The authors then show that the transport defects are specific to sorting endosomes and not acidic lysosomes (Lysotracker+) or mitochondria-this would be expected since those compartments do not have much Rab7 (please state this). (R1)

As stated by the Reviewer, mitochondria are not commonly decorated by Rab7 on their outer membrane, and despite the existence of mitochondria-lysosome contact sites (Wong et al., 2018), mitochondria are mostly devoid of Rab7. As a consequence, it is predicted that their axonal transport would not be affected by TBK1 knockdown or Rab7 S72E expression, unless these manipulations are altering trafficking at a global scale. This is precisely the reason behind these experiments. We understand that the rationale of the experimental strategy might have required further clarification and we apologise for the lack of clarity. To address this crucial point, we have changed the discussion accordingly (last paragraph on **page 14**).

(B) The system used does not make the MNs rely on signaling endosomes for support, as they would if trophic factors were exclusively presented to distal axons and were being transported in this way. (R2)

We agree with the Reviewer about the observation that in our mass culture set up, both the somatodendritic and the axonal compartment are exposed to neurotrophins and growth factors present in the motor neuron media (CNTF, GDNF, BDNF). This uniform availability might mask the requirements for neurotrophic factors being transported from the distal axon to the soma. We have added a sentence in the discussion at the end of **page 17** addressing this point, as well as stating that compartmentalised culture models could be used in future experiments.

(C) If the S72E mutant is a true phospho-mimetic mutation, it should also show higher affinity for this protein but it does not-it binds more poorly (Fig. 5E) and it shows cytoplasmic localization (as discussed by the authors). Such incorrect behavior for Rab phosphorylation mimics has been seen by others for this Rab and Rab8 and Rab10-such mutant proteins cannot be used to reflect {plus minus}phosphorylation states. Because the data indicate strongly that these are non-functional proteins, interpreting the consequences of their expression is not straightforward nor helpful. (R1)

We thank the Reviewer for highlighting the discrepancy between Rab7 p-S72 and Rab7 S72E. We had initially discussed that their different behaviour indicates Rab7 S72E is not a true phospho-mimetic mutant, but a loss-of-function, affecting p-Rab7-related processes. To address this point, we changed the revised manuscript by mentioning the abnormal cytosolic localisation and the observation that equivalent residues in Rab8 and Rab10 (T72 and T73 respectively) also turns these Rabs into loss-of-function mutants (**page 13**). We also state that despite these shortcomings, Rab7 S72E has been used as a tool to identify new Rab7 interactors, such as folliculin (Heo et al., 2018).

5. Addressing of all the minor comments from the reviewers (missing citations and correcting accuracy of statements):

Reviewer 1

Page 5 line 12. "Rab7 GTPase activity" is misleading as hydrolysis is not needed...it would be more correct to state: GTP-bound Rab7 is needed

As requested, we have changed the text to reflect that Rab7-GTP is required for the axonal transport of signalling endosomes in motor neurons.

Page 6. Line 9. If the culture is contaminated with glial cells, they would also contribute to the western blot. More likely is the use of a non-linear western blot method that exaggerates signal to noise--fluorescent antibodies give a more linear and reliable response than HRP amplification.

We completely agree with this Reviewer that glial cells also contribute to the signal in western blotting. This explains why when TBK1 knockdown is measured by immunofluorescence in motor neurons, the knockdown efficiency is lower than when measured by western blotting, which comprises all cell present in the culture. Because the way of conveying our reasoning was however not the clearest, we have modified the Results section (**page 5**) stating that TBK1 knockdown was quantified by immunofluorescence exclusively in motor neurons, whilst the second measurement was performed by western blotting including glial cells. Lastly, we also stated that we used embryonic ventral spinal cord primary cultures, which contain non-neuronal cells, not as contaminants, but as part of the diversity of cell types present in the tissue dissected.

Page 12. Line 4. Not clear how this result " suggest S72 phosphorylation contributes to membrane targeting and localization of Rab7"? PhosphoRabs are a small fraction of total Rabs in cells, and phosphorylation happens after targeting and localization.

We thank the Reviewer for this comment. We have amended the text to state that Rab7 p-S72 may help to keep a population of membrane-associated Rab7 on the target membrane for longer, instead of helping with targeting Rab7 to the membrane.

Reviewer 2

1. The authors indicate that 14.2% of HcT-positive endosome also carry TBK1 and Rab7. It would be interesting to provide the number for all the colocalization analysis of these three markers.

Following the request from this Reviewer, we have included the colocalisation analysis for all marker pairs. The percentage of organelles carrying Rab7/HcT and Rab7/TBK1; HcT/TBK1 and HcT/Rab7; and TBK1/Rab7 and TBK1/HcT are shown in **Supplementary Figure 4E**.

2. The authors state "These results show TBK1 ensures the unidirectional transport of neurotrophic signalling endosomes, through Rab7 S2 phosphorylation and the recruitment of cytoplasmic dynein. However, the data do not show any evidence of recruitment of dynein.

As pointed out by this Reviewer, we have amended the text to remove the claim about recruitment of dynein, mentioning instead the differential recruitment of RILP by Rab7 when phosphorylated.

January 5, 2026

Re: Life Science Alliance manuscript #LSA-2025-03527-T

Prof. Giampietro Schiavo
University College London
UCL Queen Square Institute of Neurology
Dept. of Neuromuscular Diseases
Queen Square
London WC1N 3BG
United Kingdom

Dear Dr. Schiavo,

Thank you for submitting your revised manuscript entitled "TBK1 activity regulates the directionality of axonal transport of signalling endosomes" to Life Science Alliance. The manuscript has been seen by the original Reviewer 1 from another journal whose comments are appended below. We also note the resolution of additional reviewer comments as discussed in our prior correspondence. While Reviewer 1 acknowledges that several issues have been resolved in the revised manuscript, they highlight important conceptual concerns.

In particular, this reviewer notes the effects of the S72A mutation diverge significantly from those of S72E. We concur this is an important discrepancy that must be directly confronted in the text, with appropriate framing in the results section as well as the discussion. The fact that serine 72 is important for Rab7 function is clear, and it is not necessarily surprising that alanine vs glutamic acid may alter the functions of Rab7 in a manner that remains to be resolved. The text must be more transparent on this point, include the motivation to examine both S72A and S72E, and must amend conclusions accordingly. Finally, this reviewer also notes that wt Rab7 is present in some key assays, a limitation must also be acknowledged.

Provided the main claims of this paper are clearly supported, and the text is transparent about what can be concluded on the molecular mechanism at play, additional data are not required in order to proceed towards publication of this work. Our general policy is that papers are considered through only one revision cycle; however, given that the suggested changes are relatively minor, we are open to one additional short round of revision. Please note that I will expect to make a final decision without additional reviewer input upon re-submission.

Please submit the final revision within one month, along with a letter that includes a point by point response to the remaining reviewer comments.

To upload the revised version of your manuscript, please log in to your account: <https://lsa.msubmit.net/cgi-bin/main.plex>
You will be guided to complete the submission of your revised manuscript and to fill in all necessary information.

B. MANUSCRIPT ORGANIZATION AND FORMATTING:

Sincerely,

Reviewer #1 (Comments to the Authors (Required)):

The manuscript contains a great deal of work and they have tried to address many concerns but the main conclusion doesn't make sense. The authors would like to conclude that TBK1 phosphorylation of Rab7 is important for a particular class of organelle motility. If that is true, why is the non-phosphorylatable S72A behaving like wild type? They claim S72A still binds RILP which is fine but it cannot be phosphorylated (and may bind RILP the same as wild type Rab7). So yes, TBK1 Knockdown has a consequence but they have not shown this goes via Rab7 phosphorylation. Also, cells still have lots of Rab7 wild type so S72E must have a dominant negative effect of undetermined nature (not discussed)? On page 11 they state that PTEN inhibition increases membrane bound Rab7 and that phosphorylation status could cause this but it is phosphoRabs that are longer lived on organelles. Thus, the authors need to be very precise in what they conclude here--the current conclusions are overstated. And this reviewer is left confused.

Response to Reviewer

Dear Dr. Schiavo,

Thank you for submitting your revised manuscript entitled "TBK1 activity regulates the directionality of axonal transport of signalling endosomes" to Life Science Alliance. The manuscript has been seen by the original Reviewer 1 from another journal whose comments are appended below. We also note the resolution of additional reviewer comments as discussed in our prior correspondence. While Reviewer 1 acknowledges that several issues have been resolved in the revised manuscript, they highlight important conceptual concerns.

In particular, this reviewer notes the effects of the S72A mutation diverge significantly from those of S72E. We concur this is an important discrepancy that must be directly confronted in the text, with appropriate framing in the results section as well as the discussion. The fact that serine 72 is important for Rab7 function is clear, and it is not necessarily surprising that alanine vs glutamic acid may alter the functions of Rab7 in a manner that remains to be resolved. The text must be more transparent on this point, include the motivation to examine both S72A and S72E, and must amend conclusions accordingly. Finally, this reviewer also notes that wt Rab7 is present in some key assays, a limitation must also be acknowledged.

Provided the main claims of this paper are clearly supported, and the text is transparent about what can be concluded on the molecular mechanism at play, additional data are not required in order to proceed towards publication of this work. Our general policy is that papers are considered through only one revision cycle; however, given that the suggested changes are relatively minor, we are open to one additional short round of revision. Please note that I will expect to make a final decision without additional reviewer input upon re-submission. Please submit the final revision within one month, along with a letter that includes a point by point response to the remaining reviewer comments.

We thank the Editor for the careful analysis of our revised manuscript and for the possibility to add further transparency to the text in line with the suggestions of Reviewer 1. The specific changes to the text are listed in our response to this Reviewer (see below).

Reviewer 1.

The manuscript contains a great deal of work and they have tried to address many concerns but the main conclusion doesn't make sense. The authors would like to conclude that TBK1 phosphorylation of Rab7 is important for a particular class of organelle motility. If that is true, why is the non-phosphorylatable S72A behaving like wild type? They claim S72A still binds RILP which is fine but it cannot be phosphorylated (and may bind RILP the same as wild type Rab7). So yes, TBK1 Knockdown has a consequence but they have not shown this goes via Rab7 phosphorylation. Also, cells still have lots of Rab7 wild type so S72E must have a dominant negative effect of undetermined nature (not discussed)? On page 11 they state that PTEN inhibition increases membrane bound Rab7 and that phosphorylation status could cause this but it is phosphoRabs that are longer lived on organelles. Thus, the authors need to be very precise in what they conclude here--the current conclusions are overstated. And this reviewer is left confused.

We thank the Reviewer for their thorough analysis of our results and conclusions. We agree that some residual conceptual misunderstanding may still affect the text. Following the Reviewer's request, we have edited our manuscript to address these issues. The main changes are indicated below.

TBK1 consensus sequence for substrate recognition is weak. As such, the regulation of the specificity of TBK1 activity relies on the binding of its adaptors, with the concomitant trans-

autoactivation and recruitment to different subcellular domains (Ma et al., 2012; Paul et al., 2025). Therefore, TBK1 might phosphorylate additional substrates on signalling endosomes. We hypothesise that this could account for the inhibition of anterograde transport, which persists upon Rab7 S72A expression, but it is relieved after TBK1 knockdown. In this way, enhanced RILP binding to Rab7 p-S72 ensures retrograde movement, but it is likely not the only way to modulate the tug-of-war taking place by opposing motors. We have acknowledged this possibility in the Result section (Page 6) and in the discussion (Page 15-16). We have also toned down our conclusions to state that RILP recruitment by Rab7 in signalling endosomes is only part of the mechanism (Page 13-14 and Page 18).

Regarding the nature of the Rab7 S72E mutation, we agree (and stated numerous times in the text) that it is not a phospho-mimetic mutant. We have also commented on the point raised by the Reviewer that Rab7 S72E behaves as a partial dominant-negative mutant, since Rab7 S72E expression is able to alter transport directionality despite endogenous Rab7 WT still being present in motor neurons (Page 13-14). Furthermore, we acknowledged the limitations linked to the presence of endogenous Rab7 in our motor neuron cultures and its impact on the conclusions of the S72E experiment (Page 13-14).

As stated by the Reviewer, phospho-Rabs are long-lived on the surface of intracellular organelles. In agreement with this, inhibiting PTEN, the phosphatase acting on Rab7 S72, increases the levels of Rab7 p-S72, causing a boost in Rab7 signal on endosomes. We have added a sentence to clarify that this type of phosphorylation on Rab GTPases switch II domain impinges on their binding to GDI and their membrane/cytosol distribution (Steger et al., 2016) (Page 11).

References

- Ma, X., E. Helgason, Q.T. Phung, C.L. Quan, R.S. Iyer, M.W. Lee, K.K. Bowman, M.A. Starovasnik, and E.C. Dueber. 2012. Molecular basis of Tank-binding kinase 1 activation by transautophosphorylation. *Proc. Natl. Acad. Sci. U.S.A.* 109:9378–9383. doi:10.1073/pnas.1121552109.
- Paul, S., S.R. Biswas, J.P. Milner, P.L. Tomsick, and A.M. Pickrell. 2025. Adaptor-Mediated Trafficking of Tank Binding Kinase 1 During Diverse Cellular Processes. *Traffic*. 26:e70000. doi:10.1111/tra.70000.
- Steger, M., F. Tonelli, G. Ito, P. Davies, M. Trost, M. Vetter, S. Wachter, E. Lorentzen, G. Duddy, S. Wilson, M.A. Baptista, B.K. Fiske, M.J. Fell, J.A. Morrow, A.D. Reith, D.R. Alessi, and M. Mann. 2016. Phosphoproteomics reveals that Parkinson's disease kinase LRRK2 regulates a subset of Rab GTPases. *Elife*. 5. doi:10.7554/eLife.12813.

January 15, 2026

RE: Life Science Alliance Manuscript #LSA-2025-03527-TR

Prof. Giampietro Schiavo
University College London
UCL Queen Square Institute of Neurology
Dept. of Neuromuscular Diseases
Queen Square
London WC1N 3BG
United Kingdom

Dear Dr. Schiavo,

Thank you for submitting your revised manuscript entitled "Introducing Life Science Alliance". The revised manuscript text removes potential reader confusion concerning the Rab S72A vs S72E mutants, and directly remarks on the discordant findings with the S72A mutant. We appreciate your understanding that these points required clarification prior to proceeding. We would be happy to publish your paper in Life Science Alliance pending final revisions necessary to meet our formatting guidelines.

MANUSCRIPT ORGANIZATION AND FORMATTING:

To avoid unnecessary delays in the acceptance and publication of your paper, please read the following information carefully. Full guidelines are available on our Instructions for Authors page, <https://www.life-science-alliance.org/authors>

- Please be sure that the authorship listing and order is correct.
- Please add the X and Bluesky handles of your host institute/organization, as well as your own, and/or one of the authors, in our system.
- Please remove supplementary information from the main manuscript text.
- Supplementary figure legends should be provided after the legends for the main figures in the manuscript file.
- LSA does not permit citation of "data not shown," "manuscript in preparation," "manuscript submitted," etc., in any section of the manuscript. Please amend the phrase on Line 283 or include the data referenced.
- Please add a Conflict of Interest statement to your main manuscript text.
- Please use the [10 author names et al.] format in your references (i.e., limit the author names to the first 10).
- Please add a callout for Figure S4B to your main manuscript text.

LSA encourages authors to provide a 30-60 second video where the study is briefly explained. We will use these videos on social media to promote the published paper and the presenting author (for examples, see <https://docs.google.com/document/d/1-UWcfbE4pGcDdcgzcmiuJl2XMBJnxKYeqRvLLrLS08s/edit?usp=sharing>). Corresponding or first-authors are welcome to submit the video. Please submit only one video per manuscript. The video can be emailed to contact@life-science-alliance.org

FINAL FILES:

The following items are required for acceptance.

The license to publish form must be signed before your manuscript can be sent to production. A link to the license to publish form will be available to the corresponding author only. Please take a moment to check your funder requirements.

Thank you for your attention to these final processing requirements. Please revise and format the manuscript and upload materials as soon as you are able.

Thank you for this interesting contribution to the literature. We look forward to publishing your paper in Life Science Alliance.

Sincerely,

January 21, 2026

RE: Life Science Alliance Manuscript #LSA-2025-03527-TRR

Prof. Giampietro Schiavo
University College London
UCL Queen Square Institute of Neurology
Dept. of Neuromuscular Diseases
Queen Square
London WC1N 3BG
United Kingdom

Dear Dr. Schiavo,

Thank you for submitting your Research Article entitled "TBK1 activity regulates the directionality of axonal transport of signalling endosomes". It is a pleasure to let you know that your manuscript is now accepted for publication in Life Science Alliance. Congratulations on this interesting work. Please forgive our error in our prior decision letter that referenced an incorrect manuscript title.

Your manuscript will now progress through copyediting and proofing. During the proofing process, please rename the Supplementary Information section to Supplementary Figure Legends. It is journal policy that authors provide original data upon request.

DISTRIBUTION OF MATERIALS:

Again, congratulations on a very nice paper and for your prompt and thorough responses to concerns by the editors and reviewers. I hope you found the review process to be constructive and are pleased with how the manuscript was handled editorially. We look forward to future exciting submissions from your lab.

Sincerely,
